# Enhancing Learning with Noisy Labels via Rockafellian Relaxation

**Louis L Chen & Eric Eckstrand**
Department of Operations Research
Naval Postgraduate School
Monterey, CA 93943, USA
`{louis.chen,eric.eckstrand}@nps.edu`

**Bobbie Chern**
Meta
Menlo Park, CA 94025, USA
`bgchern@meta.com`

**Johannes Royset**
Department of Industrial and Systems Engineering
University of Southern California
Los Angeles, CA 90089, USA
`royset@usc.edu`

**Amogh Mahapatra**
`lucky5devilhere@gmail.com`

## Abstract

Labeling errors in datasets are common, arising in a variety of contexts, such as human labeling and weak labeling. Although neural networks (NNs) can tolerate modest amounts of these errors, their performance degrades substantially once the label error rate exceeds a certain threshold. We propose the Rockafellian Relaxation Method (RRM) – an architecture-independent, loss reweighting approach to enhance the capacity of neural network methods to accommodate noisy labeled data. More precisely, it functions as a wrapper, modifying any methodology's training loss - particularly, the supervised component. Experiments indicate RRM can provide an increase to accuracy across classification tasks in computer vision and natural language processing (sentiment analysis). This observed potential for increase holds irrespective of dataset size, noise generation (synthetic/human), data domain, and adversarial perturbation.

## 1 Introduction

Labeling errors are systematic in practice, stemming from various sources. For example, the reliability of human-generated labels can be negatively impacted by incomplete information, or the subjectivity of the labeling task – as is commonly seen in medical contexts, in which experts can often disagree on matters such as the location of electrocardiogram signal boundaries Frenay & Verleysen (2014), prostate tumor region delineation, and tumor grading (Nir et al., 2018). As well, labeling systems, such as Mechanical Turk[1] often find expert labelers being replaced with unreliable non-experts (Snow et al., 2008). The impact of these errors on classification performance can be severe for machine learning approaches (Zhu & Wu, 2004), including, most notably, NNs (Krause et al., 2016; Zhang et al., 2017).

In this paper, we propose a loss-reweighting methodology for the task of training a classifier on datasets with labeling errors. More precisely, we provide a meta-algorithm that may "wrap" a loss-minimization methodology to enhance performance. The method relates to optimistic and robust distributional optimization formulations aimed at addressing adversarial training (AT), which underscores our numerical experiments on NNs that suggest this method of training can provide test performance robust to high levels of labeling error, and to some extent, feature perturbation as well. Overall, we tackle the prevalent challenge of label contamination in training data, which is a critical obstacle for deploying robust machine learning models. The approach utilizes Rockafellian Relaxations (Royset et al., 2023) in addressing contaminated labels without the need for clean validation sets or sophisticated hyper-parameters - common constraints of current methodologies.

---

[1]http://mturk.com

This distinct capability represents a key contribution, making our approach more practical for handling industrial datasets.

We discuss related works and our contributions in Section 2. In Section 3 we discuss our methodology and provide theoretical justifications. We conclude with numerical experiments/results in section 4.

## 2 RELATED WORK

In *loss adjustment* methods, individual training example losses are typically adjusted multiple times throughout the training process prior to NN updates. These methods can be further grouped into loss correction, loss reweighting, label refurbishment, and meta-learning (Song et al., 2020). Our approach has some resemblance to *loss reweighting* methods wherein each training example is assigned a unique weight, where smaller weights are assigned to examples that have likely been contaminated. For example, Ren et al. (2018); Shu et al. (2019) learn sample weights through the use of a noise-free validation set. Chang et al. (2017) assign sample weights based on prediction variances, and Zhang et al. (2021) examine the structural relationship among labels to assign sample weights. However, we view the need by these methods for a clean dataset, or at least one with sufficient class balance, as a shortcoming, and our method, in contrast, makes no assumption on the availability of such a dataset.

Recent state-of-the-art methods explore training-time interventions based on sample selection Wang et al. (2023), training dynamics, and hybrid semi-supervised methods. Xia et al. (2023) select high-discrepancy samples based on disagreement between multiple models. Kim et al. (2021) use feature-space neighborhood consistency to identify those likely to be informative/uncorrupted. Bai et al. (2021) analyze early stopping in noisy settings, arguing for patience-based mechanisms balancing memorization and underfitting. Nishi et al. (2021) use data augmentations for robustness and find that stronger augmentations, when paired with consistency objectives, significantly improve generalization in noisy-label regimes. ProMix (Xiao et al., 2023), DivideMix (Li et al., 2020) and CC (Zhao et al., 2022), treats noisy-label training as a semi-supervised learning problem, combining loss modeling with consistency regularization. Wang et al. (2022) propose scalable penalized regression to detect noise using sparsity-aware optimization, offering a fully unsupervised and efficient alternative to traditional clean-set-dependent approaches. While many of these methods achieve state-of-the-art, we show that some of their performances can be enhanced by using Rockafellian Relaxation during training.

There exist many other approaches to mitigate the effect of label noise (Song et al., 2020; Frenay & Verleysen, 2014), and can be categorized into: (1) robust architectures (Lee et al., 2019; Han et al., 2018; Goldberger & Ben-Reuven, 2017; Sukhbaatar et al., 2015; Chen & Gupta, 2015); (2) robust regularization (Shorten & Khoshgoftaar, 2019; Krogh & Hertz, 1991; Srivastava et al., 2014; Ioffe & Szegedy, 2015); and (3) robust loss functions. Robust loss functions including robust mean absolute error (MAE) (Ghosh et al., 2017), early learning regularization (ELR) (Liu et al., 2020), and generalized cross entropy (GCE) (Zhang & Sabuncu, 2018) are more robust than categorical cross entropy (CCE); however, our method is not dependent on a particular loss function, and it is possible that arbitrary loss functions, including robust MAE and ELR, can be wrapped by our method with ease.

## 3 METHODOLOGY

### 3.1 MISLABELING

Let $\mathcal{X}$ denote a *feature* space, with $\mathcal{Y}$ a corresponding *label* space. Then $\mathcal{Z} := \mathcal{X} \times \mathcal{Y}$ will be a collection of feature-label pairs, with an unknown probability distribution $D$. Throughout the forthcoming discussions, $\{(x_i, y_i)\}_{i=1}^{N}$ will denote a sample of $N$ feature-label pairs, for which some pairs will be mislabeled. More precisely, we begin with a collection $(x_i, \tilde{y}_i)$ drawn i.i.d. from $D$, but there is some unknown set $\mathcal{C} \subsetneq \{1, \ldots, N\}$ denoting (contaminated) indices for which $y_i = \tilde{y}_i$ if and only if $i \notin \mathcal{C}$. For indices $i \in \mathcal{C}$, $y_i$ is some incorrect label - the particular choice of noise/contamination matters not. Indeed, we experiment with human-generated, as well as uniform label noise in Section 4. We also consider non-uniform noise in Appendix Section D.

## 3.2 Rockafellian Relaxation Method (RRM)

Let there be given an NN architecture, for which when the parameter $\theta$ (to be optimized/learned) is set, takes as input any feature $x$ and outputs a prediction $\hat{y}$. Towards optimizing the choice of $\theta$, a prediction loss $J(\theta; x, y)$ is employed to evaluate the prediction $\hat{y}$ with respect to $y$. More precisely, training typically (e.g., Li et al. (2020); Xiao et al. (2023); Zhao et al. (2022)) involves assembling, from (noisily) labeled data, a loss $\mathcal{L}(\theta)$ given by

$$\mathcal{L}(\theta) := \frac{1}{N} \sum_{i=1}^{N} J(\theta; x_i, y_i) + r(\theta), \tag{1}$$

where a regularization term $r(\theta)$ is added to, effectively, an expected loss in which each member $i$ is drawn with probability $p_i = 1/N$. Without loss of generality, we will proceed by assuming that $r(\theta) = 0$; otherwise, any prediction loss $J(\theta; x, y)$ could be redefined as $J(\theta; x, y) + r(\theta)$.

However, if we have noisily-labeled data, we might desire to remove those members that are affected; in other words, if $\mathcal{C} \subsetneq \{1, ..., N\}$ is the set of contaminated observations, then we might desire instead to have $p_i = 0$ when $i \in \mathcal{C}$, and $p_i = \frac{1}{N-|\mathcal{C}|}$ otherwise. In this work, we consider modification of those methods that optimize a loss $\mathcal{L}$ resembling equation 1, with the intention of aligning the $p_i$ values closer to these desired (but unknown) $p$. We call this procedure the *Rockafellian Relaxation Method* (RRM), for its use of a Rockafellian Relaxation (Royset et al., 2023). More precisely, with $\Delta(N) \subseteq \mathbb{R}^N$ denoting the standard probability simplex, and $p^N$ denoting the uniformly weighted (empirical) distribution $\{p_i^N\}_{i \in [N]} = \{1/N\}$, we consider the problem

$$\min_{\theta} \quad \underbrace{\min_{p \in \Delta(N)} \mathbb{E}_{(x,y) \sim p} \left[ J(\theta; x, y) \right] + \gamma \cdot d_{TV}(p^N, p)}_{\parallel} \tag{2}$$

$$\mathcal{L}_{RRM}(\theta) := \min_{u \in U} \sum_{i=1}^{N} \left( \frac{1}{N} + u_i \right) \cdot J(\theta; x_i, y_i) + \frac{\gamma}{2} \|u\|_1,$$

where $U := \{u \in \mathbb{R}^N : \sum_{i=1}^{N} u_i = 0, \frac{1}{N} + u_i \geq 0 \ \forall i = 1, \ldots, N\}$, $\gamma > 0$, and $d_{TV}(p^N, p) := \frac{1}{2} \sum_i |p_i^N - p_i|$.

In words, given a method that optimizes some training loss $\mathcal{L}$, we can "wrap" this method with the RRM procedure by instead optimizing $\mathcal{L}_{RRM}$. In doing so, in addition to $\theta$, we obtain via equation 2 an alternative distribution $p$ that may now replace the empirical, at the cost of $\gamma$ per unit of *total variation*. Equivalently, any such alternative distribution is a loss-reweighting $u \in U$; indeed, any $u \in U$ yields a $p^u \in \Delta(N)$ via $p_i^u := 1/N + u_i$, so that $\frac{1}{2} \|u\|_1$ is the total variation distance between the empirical distribution and $p^u$.

## 3.3 Analysis and Interpretation of Rockafellian Relaxation

Although problem (2) is nonconvex in general, it is amenable to a block coordinate descent approach that iteratively cycles between updating $\theta$ and $u$. Towards this, we begin by demonstrating that the computation of $\mathcal{L}_{RRM}(\theta)$ for any fixed $\theta$, i.e., the inner-minimization (over $u \in U$) in equation 2, amounts to a tractable linear program. The following result characterizes the complete set of solutions to this linear program, and in doing so, provides an interpretation of the role that $\gamma$ plays in the loss-reweighting action of RRM.

**Theorem 3.1.** *Let $\gamma > 0$ and $\{c_i = J(\theta; x_i, y_i)\}_{i=1}^{N}$ for some $\theta$, with $c_{min} := \min_i c_i$, and $c_{max} := \max_i c_i$. Write $I_{min} := \{i : c_i = c_{min}\}$, $I_{mid} := \{i : c_i \in (c_{min}, c_{min} + \gamma)\}$, $I_{big} := \{i : c_i = c_{min} + \gamma\}$, and for any $S_1 \subseteq I_{min}, S_2 \subseteq I_{big}$, define the polytope*

$$U_{S_1, S_2}^* := \left\{ u^* : \begin{array}{c} u^* \in U \\ u_i^* \geq 0 \ \forall i \in I_{min}, \quad u_i^* = 0 \ \forall i \in S_1 \cup I_{mid}, \quad u_i^* \leq 0 \ \forall i \in I_{big} \\ u_i^* = -\frac{1}{N} \ \forall i \in S_2, \quad u_i^* = -\frac{1}{N} \ \forall i : c_i > c_{min} + \gamma. \end{array} \right\}.$$

*Then*

$$\arg\min_{u \in U} \sum_{i=1}^{N} \left( \frac{1}{N} + u_i \right) \cdot c_i + \frac{\gamma}{2} \|u\|_1 = \text{conv} \left( \bigcup_{S_1, S_2} U_{S_1, S_2}^* \right). \tag{3}$$

The theorem explains that the construction of any optimal solution $u^*$ essentially reduces to categorizing each of the losses among $\{c_i = J(\theta; x_i, y_i)\}_{i=1}^N$ as "small" or "big", according to their position in the partitioning of $[c_{min}, \infty) = [c_{min}, c_{min} + \gamma) \cup [c_{min} + \gamma, \infty)$. For losses that occur at the break points of $c_{min}$ and $c_{min} + \gamma$, this classification can be arbitrary - hence, the use of $S_1$ and $S_2$ set configurations to capture this degree of freedom. In particular, those points with losses $c_i$ exceeding $c_{min} + \gamma$ are down-weighted to zero and effectively removed from the dataset. And in the event that $c_{max} - c_{min} < \gamma$, no loss reweighting occurs.

Although Theorem 3.1 establishes that there are a plethora of valid loss-reweightings, one particular solution is intuitive, simple (efficient) to execute, and offers tunable control over the reweighting. It is obtained by setting $S_1 = S_2 = \emptyset$ in Theorem 3.1.

**Corollary 3.1.1.** *Let $\gamma > 0$ and $\theta$ be given, yielding losses $\{c_i = J(\theta; x_i, y_i)\}_{i=1}^N$. Define $\chi(\theta) := \{i : c_i > c_{min} + \gamma\}$ and let $u^*(\theta)$ given by $\{u_i^*(\theta)\}_{i \in I_{min}} = \{\frac{|\chi(\theta)|}{N \cdot |I_{min}|}\}$, $\{u_i^*(\theta)\}_{i \in I_{mid} \cup I_{big}} = \{0\}$, and $u_i^*(\theta) = \frac{-1}{N}$ for all $i \in \chi(\theta)$. Then $u^*(\theta)$ solves the linear program $\mathcal{L}_{RRM}(\theta)$, equiv., the inner minimization of equation 2.*

In words, given any $\theta$, there is always an optimal reweighting (for problem $\mathcal{L}_{RRM}(\theta)$) in which $|\chi(\theta)|$ - many samples are removed from training; specifically, it transfers $\frac{|\chi(\theta)|}{N}$ probability mass away from the group of highest-cost examples and uniformly re-distributes among $I_{min}$. Further, as $\gamma$ crucially determines $\chi(\theta)$, we see that by tuning $\gamma$ we can control the fraction of samples that are pruned, a method that we expand upon in the forthcoming Section 3.5.1, and for which we executed in many experiments of Section 4.1.

If over the course of an iterative algorithmic scheme, e.g., Algorithm 1, $\chi(\theta)$ converges to some set, then this set is effectively removed even if the training of $\theta$ might proceed. The experiments of Section 4.2.2 (see Table 6) display such convergent behavior.

## 3.4 RRM and Optimistic Wasserstein Distributionally Robust Optimization

In this section, we discuss RRM's relation to distributionally robust and optimistic optimization formulations. Indeed, (2)'s formulation as a min-min problem bears resemblance to optimistic formulations of recent works, e.g., Nguyen et al. (2019). We will see as well that the minimization in $u$, as considered in Theorem 3.1, relates to an approximation of a data-driven Wasserstein Distributionally Robust Optimization (DRO) formulation (Staib & Jegelka, 2017).

### 3.4.1 Loss-reweighting via Data-Driven Wasserstein Formulation

For this discussion, as it relates to reweighting, we will lift the feature-label space $\mathcal{Z} = \mathcal{X} \times \mathcal{Y}$. More precisely, we let $\mathcal{W} := \mathbb{R}_+$ denote a space of *weights*. Next, we say $\mathcal{W} \times \mathcal{Z}$ has an unknown probability distribution $\mathcal{D}$ such that $\pi_{\mathcal{Z}}\mathcal{D} = D$ and $\Pi_{\mathcal{W}}\mathcal{D}(\{1\}) = 1$. In words, all possible (w.r.t. $D$) feature-label pairs have a weight of 1. Finally, we define an *auxiliary loss* $\ell : \mathcal{W} \times \mathcal{Z} \times \Theta$ by $\ell(w, z; \theta) := w \cdot J(x, y; \theta)$, for any $z = (x, y) \in \mathcal{Z}$.

Given a sample $\{(1, x_i, y_i)\}_{i=1}^N$, just as in Section 3.2, we can opt not to take as granted the resulting empirical distribution $\mathcal{D}_N$ because of the possibility that $|\mathcal{C}|$-many have incorrect labels (i.e., $y_i \neq \tilde{y}_i$). Instead, we will admit alternative distributions obtained by shifting the $\mathcal{D}_N$'s probability mass off "contaminated" tuples $(1, x_i, y_i)_{i \in \mathcal{C}}$ to possibly $(0, x_i, y_i)$, $(1, x_i, \tilde{y}_i)$, or even some other tuple $(1, x_j, \tilde{y}_j)$ with $j \notin \mathcal{C}$ for example - equivalently, eliminating, correcting, or replacing the sample, respectively. In order to admit such favorable corrections to $\mathcal{D}_N$, we can consider the optimistic (Nguyen et al., 2019; Staib & Jegelka, 2017) data-driven problem

$$\min_{\theta} \left( v_N(\theta) := \min_{\tilde{\mathcal{D}}: W_1(\mathcal{D}_N, \tilde{\mathcal{D}}) \leq \epsilon} \mathbb{E}_{\tilde{\mathcal{D}}} [\ell(w, z; \theta)] \right), \tag{4}$$

in which for each parameter tuning $\theta$, $v_N(\theta)$ measures the expected auxiliary loss with respect to the most favorable distribution within an $\epsilon$-prescribed $W_1$ (1-Wasserstein) distance of $\mathcal{D}_N$. It turns out that a budgeted deviation of the weights alone (and not the feature-label pairs) can approximate (up to an error diminishing in $N$) $v_N(\theta)$. More precisely, we derive the following approximation along similar lines to Staib & Jegelka (2017).

**Proposition 3.1.** *Let $\epsilon > 0$, and suppose for any $\theta$, $\max_{(x,y)\in\mathcal{Z}} |J(\theta; x, y)| < \infty$. Then there exists $\kappa \geq 0$ such that for any $\theta$, the following problem*

$$v_N^{MIX}(\theta) := \min_{u_1,\ldots,u_N} \sum_{i=1}^{N} (\frac{1}{N} + u_i) \cdot J(\theta; x_i, y_i) + \gamma_\theta \|u\|_1$$

$$s.t. \ \ u_i + \frac{1}{N} \geq 0 \ \ i = 1, \ldots, N$$

*satisfies $v_N(\theta) + \frac{\kappa}{N} \geq v_N^{MIX}(\theta) \geq v_N(\theta)$. In particular, $-\gamma_\theta \leq \min_i J(\theta; x_i, y_i)$, and, for any $u^*$ solving $v_N^{MIX}(\theta)$, it holds that $u_i^* = -\frac{1}{N}$ for $i$ such that $J(\theta; x_i, y_i) > \gamma_\theta$.*

In summary, while the optimistic Wasserstein formulation would permit correction to $\mathcal{D}_N$ with a combination of reweighting and/or feature-label revision, the above indicates that a process focused on reweighting alone could accomplish a reasonable approximation; further, upon comparison to (2), we see that RRM is a constrained version of this approximating problem, that is,

$$\mathcal{L}_{RRM}(\theta) \geq v_N^{MIX}(\theta) \geq v_N(\theta).$$

Hence, RRM is optimistic but not as much as the data-driven Wasserstein approach.

### 3.5 A-RRM/RRM ALGORITHM

Towards solving problem (2) in the two decisions $\theta$ and $u$, we proceed iteratively with a block-coordinate descent heuristic outlined in Algorithm 1, whereby we update the two separately in cyclical fashion. In other words, we update $\theta$ while holding $u$ fixed, and we update $u$ whilst holding $\theta$ fixed. The update of $\theta$ is an SGD step on a batch of $s$-many samples. The update of $u$ is a linear program, but can also be executed instantaneously via Corollary 3.1.1.

---

**Algorithm 1** (Adversarial) Rockafellian Relaxation Algorithm (A-RRM/RRM)

---

**Require:** Loss Function $J(\theta; x, y)$, training perturbation $\epsilon \in [0, 1]$, Number of epochs $\sigma$, Batch size $s \geq 1$, learning rate $\eta > 0$, threshold $\gamma > 0$, reweighting step $\mu \in (0, 1)$. (optional) contamination estimate $C' \in [0, 1]$.
  $u \leftarrow 0 \in \mathbb{R}^N$
  $Iter \leftarrow 0$
  **repeat**
    $Iter \leftarrow Iter + 1$
    $\theta \leftarrow \texttt{GradientSteps}(\sigma, s, \eta, \epsilon, \theta, u)$
    **if** $C'$ provided **then**
      $\ell_{1-C'} \leftarrow \max\{\ell : \frac{|\{j : J(\theta; x_j^b, y_j^b) > \ell\}|}{N} \geq C'\}$
      $\gamma \leftarrow \ell_{1-C'} - \min_i J(\theta; x_i^b, y_i^b)$
      $\mu \leftarrow 1$
    **end if**
    $u \leftarrow \texttt{Re-weight}(\gamma, \mu)$
  **until** Desired Validation Accuracy or Loss, or $Iter$ is sufficiently large
  Return $\theta$

---

Apart from the learning rate $\eta$ that is (industry) standard in gradient-based algorithms, Algorithm 1 can be parameter-less; more precisely, the step-size $\mu$ and threshold $\gamma$ parameters can in fact be auto-tuned, in a manner that follows from Corollary 3.1.1, so as to precisely control how many examples are pruned in each re-weighting step, as might be desired or guided by a contamination estimate $C'$.

### 3.5.1 AUTO-TUNING: PRECISE SAMPLE PRUNING WITH A CONTAMINATION ESTIMATE $C'$

As outlined in Algorithm 1, if an estimate $C'$ of the true contamination $C$ is on hand, then by setting $\gamma = \ell_{1-C'} - \min_i J(\theta; x_i, y_i)$, where $\ell_{1-C'}$ is approximately the $(1 - C')-$th quantile of the costs,

---

[1]A minimizer can be obtained via a linear program solver or explicitly via the solution of Corollary 3.1.1.

---

**Algorithm 2** `GradientSteps`$(\sigma, s, \eta, \epsilon, \theta, u)$

---

**Require:** $\sigma \in \mathbb{Z}_{\geq 1}, s \in \mathbb{Z}_{\geq 1}, u \in U, \eta > 0, \epsilon \in [0,1], \theta$
  **for** $e = 1, \ldots, \sigma$ **do**
    **for** $b = 1, \ldots, \lceil \frac{N}{s} \rceil$ **do**
      $\{(x_i^b, y_i^b)\}_{i=1}^s \leftarrow$ draw batch from $\{(x_i, y_i)\}_{i=1}^N$
      **for** i = 1, ..., s **do**
        $x_i^b \leftarrow x_i^b + \epsilon \cdot sign\left(\nabla_x J(\theta; (x_i^b, y_i^b))\right)$
      **end for**
      $\theta \leftarrow \theta - \eta \sum_{i=1}^s \left(\frac{1}{N} + u_i\right) \cdot \nabla_\theta J(\theta; (x_i^b, y_i^b))$
    **end for**
  **end for**
  Return $\theta$

---

**Algorithm 3** `Re-weight`$(\gamma, \mu)$

---

**Require:** threshold $\gamma > 0$, re-weighting step $\mu \in (0, 1)$.
  $u^* \leftarrow \min_{u \in U} \sum_{i=1}^N \left(\frac{1}{N} + u_i\right) \cdot J(\theta; x_i, y_i) + \gamma \|u\|_1$ [1]
  Return $u \leftarrow \mu u^* + (1 - \mu)$

---

or equivalently, $\max\{J(\theta; x_i, y_i) : \frac{\{j : J(\theta; x_j, y_j) > J(\theta; x_i, y_i)\}}{N} \geq C'\}$, then the fraction of observations $\frac{|\chi|}{N} = \frac{|\{i : J(\theta; x_i, y_i) > \gamma + \min_j J(\theta; x_j, y_j)\}|}{N}$ that the reweighting of Corollary 3.1.1 "prunes" is at least $C'$. Sections 4.1, 4.2.1 experiments implement this form of RRM, and we examine how RRM enhancement scales with the quality of the estimate $C'$ in Appendix Section B experiments.

### 3.5.2  RRM ($\epsilon = 0$) AND A-RRM ($\epsilon > 0$)

We will refer to Algorithm 1 as RRM or A-RRM, when $\epsilon = 0$ or $\epsilon > 0$, respectively. The only difference lies in the execution of (Algorithm 2's) `GradientSteps`.

When $\epsilon = 0$, `GradientSteps` simply executes $\sigma$ iterations of SGD. But if $\epsilon > 0$, then `GradientSteps` additionally executes a Fast Gradient Sign Method (FGSM) (Goodfellow et al., 2015) adversarial attack, whereby each time a batch $(x_i^b, y_i^b)_{i=1}^s$ is drawn, every $x_i^b$ is perturbed via $x_i^b + \epsilon \cdot sign(\nabla_x J(\theta, x_i^b, y_i^b))$. We note that other adversarial perturbation methods can be substituted in place of this choice of attack. Then a Stochastic gradient descent (SGD) step is taken.

### 3.5.3  ON COMPLEXITY

Each iteration of Algorithm 1 is comprised of two different tasks: 1.) `GradientSteps` and 2.) `Re-weight`. Task 1's gradient step calculations are entirely standard practice. Task 2 amounts to solving the polynomial-sized (in training data) linear program of equation 3, for which general theoretical efficiency is well-established and either a commercial solver (e.g. CPLEX 12.8.0) would suffice, or, even quicker, the solution of Corollary 3.1.1 could be derived with a single pass over the list of losses (see experiments of Sections 4.1, 4.2.1). We especially note that `Re-weight` is only executed after all the $\sigma$- many epochs of `GradientSteps` have concluded. In summary, the scaling of computation with larger datasets is not limiting in nature. We refer the reader to Section 4.2.1 and Appendix Section C for experiments on MNIST-3 and CIFAR-10 with average computation times of Algorithm 3 (`Re-weight`) reported.

## 4  EXPERIMENTS AND RESULTS

### 4.1  RRM ENHANCEMENT IN REAL-WORLD NOISE EXPERIMENTS

We experimented on the CIFAR-N (Wei et al., 2022), Clothing-1M (Xiao et al., 2015), and Food-101N (Lee et al., 2018) datasets which all exhibit real-world noise. Specifically, for each dataset, we identified a handful of top performing methods to wrap, and then compared the resulting performance against the leaderboard performers for that dataset.

CIFAR-10N/100N furnishes CIFAR-10/100 with human-annotated, noisy labels from Amazon Mechanical Turk, and the top two performers for this dataset are ProMix and Divide-Mix. Clothing1M is a dataset composed of 1 million images scraped from online shopping websites, with labels derived from accompanying text, and the top two performers for this dataset are LRA-diffusion and CC. Food-101N consists of 310009 noisily-labeled training images collected from the web, and the top two performers for this dataset are LRA-diffusion and SURE.

We chose to wrap ProMix, Divide-Mix, CC, and CCE to compare against the three leaderboards.

ProMix, Divide-Mix, and CC all exhibit a training loss of the form $\mathcal{L}(\theta) := \mathcal{L}_X(\theta) + r(\theta)$, where $r(\theta) := \lambda_{\mathcal{U}}\mathcal{L}_{\mathcal{U}}(\theta) + \mathcal{L}_{aux}(\theta)$. In words, the loss combines a supervised component $\mathcal{L}_X$, a ($\lambda_{\mathcal{U}}$-weighted) semi-supervised component $\lambda_{\mathcal{U}}\mathcal{L}_{\mathcal{U}}$, and some additional auxiliary loss $\mathcal{L}_{aux}$. In particular, these methods use some procedure to select a subset $X$ of labeled data for which to formulate an average loss of the form $\mathcal{L}_X(\theta) = \frac{1}{|X|} \sum_{i \in \mathcal{X}} J(\theta; x_i, y_i)$ for some classification loss $J$. While these methods endeavor for $X$ to be clean of noisily -labeled data, this cannot be guaranteed in practice.

In the wrapping of these methods by RRM, we replace the training loss $\mathcal{L}$ with equation 2's $\mathcal{L}_{RRM}$. We do so similarly for CCE with $r(\theta) = 0$. More precisely, after every $\sigma$ gradient steps (see Algorithm 2) for $\theta$, we fix the current $\theta$ and execute a reweighting by solving for $u$ (see Algorithm 3). For ProMix, Divide-Mix, and CC, $\sigma = 10$. For CCE, $\sigma = 1$. As for $\gamma$, this was set by employing Section 3.5.1's use of a conservative estimate $C'$ of label error rate 0.5, 0.5, 0.4, and 0.2 for datasets CIFAR-10N, CIFAR-100N, Clothing1M, and Food101N, respectively.

As for any hyperparameter of ProMix, Divide-Mix, or CC, in our experiments we simply re-used those reported in their respective papers' (Xiao et al., 2023; Li et al., 2020; Zhao et al., 2022) experimental setup.

**CIFAR-N Results:** Table 1 lists the top published accuracies for comparison, with asterisks marking state-of-the-art. On CIFAR-100N, we achieved state-of-the-art accuracy with both the wrappings of ProMix and Divide-Mix. On CIFAR-10N, wrapping enhanced Divide-Mix while ProMix's accuracy was virtually unaffected.

Table 1: CIFAR-N Test Accuracies.

| Method\Dataset | CIFAR-10N (Worst) | CIFAR- 100N (Noisy Fine) |
|---|---|---|
| ProMix **(+ RRM)** | 96.34* (96.32) | 73.79 (74.19*) |
| Divide-Mix **(+ RRM)** | $92.56 \pm 0.42$ ($94.75^{**} \pm 0.08$) | $71.13 \pm 0.48$ ($73.98^{**} \pm 0.05$) |
| SOP | $93.24 \pm 0.21$ | $67.87 \pm 0.23$ |
| ELR+ | $91.09 \pm 1.60$ | $66.72 \pm 0.07$ |
| Co-Teaching+ | $83.26 \pm 0.17$ | $57.88 \pm 0.24$ |
| GCE | $80.66 \pm 0.35$ | $56.73 \pm 0.30$ |

**Clothing1M and Food-101N Results:** Tables 2 and 3 list the top published accuracies by methods on Clothing1M and Food-101N respectively for comparison, with asterisks marking state-of-the-art. On Clothing1M, we enhanced a top-performer in CC to perform virtually as well as the best performing LRA-diffusion method. We also examined to what extent conventional training with CCE loss, a common baseline, could be enhanced by RRM to accommodate noisy-labeling. Along these lines, upon wrapping CCE, we found that it now outperformed some noisy label methods (JoCoR and GCE) on Clothing1M. On Food-101N, we achieved a top accuracy by wrapping the traditional CCE method.

In Tables 1,2, and 3, wherever confidence intervals are omitted, this is because they are in fact not reported by the authors of that method.

## 4.2 RRM ENHANCEMENT ACROSS LEVELS OF LABEL NOISE AND ADVERSARIAL PETURBATION

Unlike the previous section's comparisons with state-of-the-art methods, here the focus is on how RRM enhancement of various methods scales with $C$ through synthetically generated label noise experiments on CIFAR-10 (Alex, 2009) and MNIST-10 (LeCun & Cortes, 2010). We refer to

Table 2: Clothing1M Test Accuracies.

| Method\Dataset | Clothing1M |
|---|---|
| CC (+ RRM) | 75.4 (75.69**) |
| CCE (+ RRM) | 68.94 (71.48 $\pm$ 0.25) |
| LRA-diffusion | 75.7* |
| ELR+ | 74.81 |
| DivideMix | 74.76 |
| Knockoffs-SPR | 75.2 |
| Meta-Weight-Net | 73.72 |

Table 3: Food-101N Test Accuracies.

| Method\Dataset | Food-101N |
|---|---|
| CCE (+RRM) | 81.67 (84.21) |
| LRA-diffusion | 93.42* |
| SURE | 88.0** |
| LongReMix | 87.39 |
| CleanNet | 83.95 |

Appendix (Section E) for further experimentation on Toxic Comments, IMDb and Tissue Necrosis datasets.

### 4.2.1 Synthetic Experiments with only Label Noise

**Implementation Details:** We employ symmetric noise at a rate of $C$ on 50000 CIFAR-10 training labels. The 10000 testing labels are unperturbed. A ResNet-34 architecture is utilized as in Zhang & Sabuncu (2018), along with their data preprocessing and augmentation scheme. Two sets of experiments are performed. As a baseline, 180 epochs are performed for CCE, MAE and MSE loss functions without RRM wrapping. Training label contamination settings of 10%, 20%, and 30% are utilized. In the other set of experiments, CCE, MAE and MSE loss functions are wrapped with RRM and executed 12 iterations with $\sigma = 15$ epochs per iteration, also for a total of 180 epochs. Stochastic gradient descent (SGD) with a learning rate ($\eta$) of 10.0, momentum of 0.9 with Nesterov momentum enabled, and a weight decay of $10^{-5}$ are employed for both sets of experiments. Each experiment is executed 5 times and the average test set accuracy and standard deviation is reported in Table 4, with the RRM results placed in parentheses.

**Analysis:** As Table 4 indicates, both CCE and MSE methods greatly benefit from an RRM approach across contamination levels, whereas the benefit of RRM for MAE is less definitive. The additional computation time incurred from the Re-weight($\gamma, \mu$) step is an average of 3.88 seconds per execution of every iteration, for a total of 46.56 seconds.

Table 4: **CIFAR-10** Test accuracy (%) comparison between CCE, MAE, and MSE loss functions across contamination levels. RRM-wrapped accuracy results are in parentheses. Values marked with ($\dagger$) are from Zhang & Sabuncu (2018). Values marked with (*) are the best.

| Method | Contamination $C$ | | |
|---|---|---|---|
| | 10% | 20% | 30% |
| CCE (+RRM) | $89.94 \pm 0.29$ ($92.20 \pm 0.26$) | $86.98 \pm 0.44^{\dagger}$ ($90.44 \pm 0.31$) | $81.90 \pm 0.86$ ($88.49 \pm 0.33$) |
| MAE (+RRM) | $92.04 \pm 0.25$ ($90.10 \pm 3.56$) | $83.72 \pm 3.84^{\dagger}$ ($87.82 \pm 4.07$) | $88.28 \pm 3.52$ ($82.98 \pm 6.55$) |
| MSE (+RRM) | $91.83 \pm 0.25$ ($93.04^* \pm 0.17$) | $89.43 \pm 0.37$ ($91.43^* \pm 0.32$) | $86.34 \pm 0.22$ ($89.95^* \pm 0.55$) |

We refer the reader to Appendix Section C for experiments with higher levels of contamination.

### 4.2.2 Synthetic Experiments with both Label Noise and Adversarial Perturbation

In the following experiment we evaluate RRM in a setting of both adversarial feature perturbation and label contamination.

**Adversarial Training (AT) Baseline:** As baseline, we execute a standard adversarial training (AT) approach, in which we execute A-RRM, omitting the Re-weight step. Indeed, it is common to perform training on data augmented with adversarial examples crafted through FGSM (Madry et al., 2018). Further, using this benchmark, we may highlight the value of re-weighting.

**Test-set Perturbation** $\epsilon_{test}$: In the experiments, upon conclusion of the A-RRM training for which a trained $\theta^*$ is obtained, we executed an FGSM attack on each member $(x, y)$ of the test set via $x + \epsilon_{test} \cdot \nabla J(\theta^*; x, y)$. In this way, we explore the harm to performance that a misalignment between our training perturbation $\epsilon$ and a true test-set perturbation $\epsilon_{test}$ could have.

**Implementation Details:** We employ symmetric noise at a rate of $C$ on 60000 MNIST-10 training labels. The 10000 testing labels are unperturbed. Twenty percent of the training data is set aside for validation purposes. We adopt a basic CNN architecture with a few convolutional layers. The first layer has a depth of 32, and the next two layers have a depth of 64. Each convolutional layer employs a kernel of size three and the ReLU activation function followed by a max-pooling layer employing a kernal of size 2. The last convolutional layer is connected to a classification head consisting of a 100-unit dense layer with ReLU activation, followed by a 10-unit dense layer with softmax activation. Categorical cross-entropy is employed for the loss function. Using Tensorflow 2.10 (Abadi et al., 2015), 50 iterations of A-RRM are executed with $\sigma = 10$ epochs per iteration for a total of 500 epochs for a given hyperparameter setting. For A-RRM, the hyperparameter settings of $\mu$ and $\gamma$ at 0.5 and 2.0, respectively, are based on a search to optimize validation set accuracy. Separately, we perform a comparable 500 epochs using AT only. Both AT and A-RRM employ stochastic gradient descent (SGD) with a learning rate ($\eta$) of 0.1. An $\epsilon = 1.0$ is used for all training image perturbations.

**Performance Analysis:** For each of the 0%, 5%, 10%, 20%, and 30% training label contamination levels, we compare adversarial training (AT) and A-RRM performance under various regimes of test set perturbation ($\epsilon_{test} \in \{0.0, 0.1, 0.25, 0.5, 1.0\}$). In Table 5 we show the test set accuracy achieved when validation set accuracy peaks. We can see that training with an $\epsilon = 1.0$ and testing with lower $\epsilon_{test}$ levels of $0.00, 0.10$, and $0.25$, results in a drastic degradation in accuracy for AT for contamination levels greater than 0%. This performance collapse is not observed when using A-RRM. Given that the $\epsilon$ employed during training may not match test/production environment ($\epsilon_{test}$), our findings suggest that A-RRM can confer a greater benefit than AT in these scenarios.

**Convergence of $u$:** Table 6 tracks the evolution of the $u_i$-values across 49 iterations of Algorithm 1 for 9600 contaminated and 38400 clean MNIST training samples at 20% contamination. After the first iteration, nearly all $u_i \approx 0$. By iteration 10, many contaminated samples already exhibit negative $u_i$-values, while most clean samples remain near zero. By iteration 49, 9286 of the 9600 contaminated samples fall in $(-2.08, -1.56] \cdot 10^{-5}$, effectively canceling their nominal probability $1/N = 2.08 \cdot 10^{-5}$, so they are pruned from training. In contrast, most clean samples (35246 of 38400) retain their nominal probability $1/N = 2.08 \cdot 10^{-5}$. This selective downweighting explains A-RRM's advantage over AT: A-RRM suppresses contaminated data in situ, a critical property when test-time perturbations are unknown or milder than those used during training. Interestingly, some $u_i$- values presented **"close-calls"**, i.e., were such that $\frac{1}{N} + u_i \approx 0$ at some iteration before then recovering by the conclusion of training to have a positive sample weight $\frac{1}{N} + u_i \geq \frac{1}{2N} > 0$, suggesting RRM's ability to recover a borderline but ultimately learnable sample that was assigned to near-zero weight early. We refer the reader to Appendix F for details.

Table 5: **MNIST-10** Test accuracy (%) for AT and A-RRM under different levels of contamination $C$, training perturbation $\epsilon = 1.0$, and test-set adversarial perturbation $\epsilon_{test}$.

| | | $C$ | | | | | | | | | |
| | | 0% | | 5% | | 10% | | 20% | | 30% | |
| | | AT | A-RRM | AT | A-RRM | AT | A-RRM | AT | A-RRM | AT | A-RRM |
|---|---|---|---|---|---|---|---|---|---|---|---|
| $\epsilon_{test}$ | 0.00 | **97** | 96 | 63 | **95** | 57 | **97** | 58 | **96** | 26 | **86** |
| | 0.10 | **95** | 93 | 64 | **92** | 71 | **94** | 61 | **93** | 20 | **82** |
| | 0.25 | **93** | 90 | 83 | **91** | 88 | **92** | 84 | **90** | 74 | **81** |
| | 0.50 | **91** | 88 | **94** | 91 | **94** | 90 | **90** | 88 | **97** | 80 |
| | 1.00 | **86** | 83 | **95** | 90 | **94** | 86 | **88** | 83 | **98** | 77 |

## 5 CONCLUSION

RRM is a novel loss reweighting procedure that enhances learning with noisy labels, as our experiments demonstrate across datasets and noise schemes. In fact, we achieve state-of-the-art performance on CIFAR-100N when wrapping ProMix and DivideMix. Further, our experiments indicate improvements that persist across model architectures and applications. The method is virtually hyperparameter-free, upon using an auto-tuning feature to control the `Re-weight` pruning action with an estimate of label contamination. Our experiments further show that RRM boosts the test accuracy of both standard and noise-robust methods, and that A-RRM shows promise in extending

Table 6: **MNIST-10** Evolution of $u$ across $|C| = 9600$ contaminated data points and $N - |C| = 38400$ clean data points. Note that $1/N = 2.08 \cdot 10^{-5}$.

| | | 1. iteration | | 10. iteration | | 49. iteration | |
|---|---|---|---|---|---|---|---|
| | | $C$ | $[N] \setminus C$ | $C$ | $[N] \setminus C$ | $C$ | $[N] \setminus C$ |
| $u_i$ value $(10^{-5})$ | $\gg 0$ | 0 | 1 | 0 | 4 | 0 | 25 |
| | $\approx 0$ | 8844 | 38385 | 2058 | 37524 | 91 | 35246 |
| | (-0.52, 0.00) | 0 | 0 | 7 | 36 | 146 | 1655 |
| | (-1.04, -0.52] | 0 | 0 | 41 | 45 | 43 | 155 |
| | (-1.56, -1.04] | 756 | 14 | 415 | 174 | 34 | 168 |
| | (-2.08, -1.56] | 0 | 0 | 7079 | 617 | 9286 | 1151 |

these gains to settings with combined adversarial perturbations and label noise, yielding models more resilient to both. For more experiments showcasing enhancement, please see the Appendix.

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

# A  APPENDIX/SUPPLEMENTAL MATERIAL

## A.1  SECTION 3 PROOFS

**Theorem 3.1.** *Let $\gamma > 0$ and $\{c_i = J(\theta; x_i, y_i)\}_{i=1}^N$ for some $\theta$, with $c_{min} := \min_i c_i$, and $c_{max} := \max_i c_i$. Write $I_{min} := \{i : c_i = c_{min}\}$, $I_{mid} := \{i : c_i \in (c_{min}, c_{min} + \gamma)\}$, $I_{big} := \{i : c_i = c_{min} + \gamma\}$, and for any $S_1 \subseteq I_{min}$, $S_2 \subseteq I_{big}$, define the polytope*

$$U^*_{S_1,S_2} := \left\{ u^* : \begin{array}{c} u^* \in U \\ u_i^* \geq 0 \ \forall i \in I_{min}, \quad u_i^* = 0 \ \forall i \in S_1 \cup I_{mid}, \quad u_i^* \leq 0 \ \forall i \in I_{big} \\ u_i^* = -\frac{1}{N} \ \forall i \in S_2, \quad u_i^* = -\frac{1}{N} \ \forall i : c_i > c_{min} + \gamma. \end{array} \right\}.$$

*Then*

$$\arg\min_{u \in U} \sum_{i=1}^N (\frac{1}{N} + u_i) \cdot c_i + \frac{\gamma}{2} \|u\|_1 = \mathrm{conv}\left( \bigcup_{S_1, S_2} U^*_{S_1, S_2} \right). \tag{3}$$

*Proof.* For any set $C$, let $\iota_C(x) = 0$ and $\iota_C(x) = \infty$ otherwise. We recognize that $u^\star$ is a solution of the minimization problem if and only if it is a minimizer of the function $h$ given by

$$h(u) = \sum_{i=1}^N \left( c_i/N + u_i c_i + \frac{\gamma}{2} |u_i| + \iota_{[0,\infty)}(1/N + u_i) \right) + \iota_{\{0\}}\left( \sum_{i=1}^N u_i \right)$$

Thus, because $h(u) > -\infty$ for all $u \in \mathbb{R}^N$ and $h$ is convex, $u^\star$ is a solution of the minimization problem if and only if $0 \in \partial h(u^\star)$. We proceed by characterizing $\partial h$.

Consider the univariate function $h_i$ given by

$$h_i(u_i) = c_i/N + u_i c_i + \frac{\gamma}{2} |u_i| + \iota_{[0,\infty)}(1/N + u_i).$$

For $u_i \geq -1/N$, the Moreau-Rockafellar sum rule reveals that

$$\partial h_i(u_i) = c_i + \begin{cases} \{\frac{\gamma}{2}\} & \text{if } u_i > 0 \\ [-\frac{\gamma}{2}, \frac{\gamma}{2}] & \text{if } u_i = 0 \\ \{-\frac{\gamma}{2}\} & \text{if } -1/N < u_i < 0 \\ (-\infty, -\frac{\gamma}{2}] & \text{if } u_i = -1/N. \end{cases}$$

For $u = (u_1, \ldots, u_N) \in [-1/N, \infty)^N$, we obtain by Proposition 4.63 in Royset & Wets (2021) that

$$\partial\left( \sum_{i=1}^N h_i \right)(u) = \partial h_1(u_1) \times \cdots \times \partial h_N(u_N).$$

Let $h_0$ be the function given by $h_0(u) = \iota_{\{0\}}(\sum_{i=1}^N u_i)$. Again invoking the Moreau-Rockafellar sum rule while recognizing that the interior of the domain of $\sum_{i=1}^N h_i$ intersects with the domain of $h_0$, we obtain

$$\partial h(u) = \partial\left( \sum_{i=1}^N h_i \right)(u) + \partial h_0(u) = \partial h_1(u_1) \times \cdots \times \partial h_N(u_N) + \begin{bmatrix} 1 \\ \vdots \\ 1 \end{bmatrix} \mathbb{R}$$

for any $u = (u_1, \ldots, u_N)$ with $u_i \geq -1/N$, $i = 1, \ldots, N$, and $\sum_{i=1}^N u_i = 0$. Hence, $u^* \in U$ is optimal if and only if for some $\lambda \in \mathbb{R}$,

$$\lambda \in \begin{cases} \{c_i + \frac{\gamma}{2}\} & \text{if } u_i^\star > 0 \\ [c_i - \frac{\gamma}{2}, c_i + \frac{\gamma}{2}] & \text{if } u_i^\star = 0 \\ \{c_i - \frac{\gamma}{2}\} & \text{if } u_i^\star \in (-1/N, 0) \\ (-\infty, c_i - \frac{\gamma}{2}] & \text{if } u_i^\star = -1/N. \end{cases} \tag{5}$$

It follows that any such $\lambda$ must then satisfy $\lambda \leq c_{\min} + \frac{\gamma}{2}$, as the above list of cases indicate for arbitrary $i$ and no matter the value of $u_i^*$, for any optimal $u^* \in U$.

We proceed to show that in fact, $u^* \in U$ solves equation 3 if and only if $\lambda = c_{min} + \frac{\gamma}{2}$ satisfies the conditions of equation 5 with respect to $u^*$ for all $i$. Suppose $u^* \in U$ is optimal and that there exists a corresponding $\lambda$ satisfying equation 5 for all $i$, and that it satisfies $\lambda < c_{min} + \frac{\gamma}{2}$. Then for any optimal $u^* \in U$, there exists no $i$ for which $u_i^* > 0$, otherwise $c_i = \lambda - \frac{\gamma}{2} < c_{min}$, an impossibility. This means any optimal $u^* \in U$ must satisfy $u^* \leq 0$, for which then we would conclude that in fact $u^* = 0$, and is the unique optimal solution, which in turn yields that $\lambda - \frac{\gamma}{2} \leq c_i \leq \lambda + \frac{\gamma}{2} < c_{min} + \gamma$ for all $i$. This last inequality reveals that $\lambda^* := c_{min} + \frac{\gamma}{2}$ is compatible with the unique optimal solution $u^* = 0 \in U$ in such a case. In summary, if a $\lambda < c_{min} + \frac{\gamma}{2}$ is compatible with all optimal solutions $u^* \in U$, then $\lambda^* := c_{min} + \frac{\gamma}{2}$ is as well.

It follows that $u^* \in U$ solves equation 3 if and only if $\lambda = c_{min} + \frac{\gamma}{2}$ satisfies the conditions of equation 5 with respect to $u_i^*$ for all $i$; hence, the result follows. $\square$

**Corollary 3.1.1.** *Let $\gamma > 0$ and $\theta$ be given, yielding losses $\{c_i = J(\theta; x_i, y_i)\}_{i=1}^N$. Define $\chi(\theta) := \{i : c_i > c_{min} + \gamma\}$ and let $u^*(\theta)$ given by $\{u_i^*(\theta)\}_{i \in I_{min}} = \{\frac{|\chi(\theta)|}{N \cdot |I_{min}|}\}$, $\{u_i^*(\theta)\}_{i \in I_{mid} \cup I_{big}} = \{0\}$, and $u_i^*(\theta) = \frac{-1}{N}$ for all $i \in \chi(\theta)$. Then $u^*(\theta)$ solves the linear program $\mathcal{L}_{RRM}(\theta)$, equiv., the inner minimization of equation 2.*

*Proof.* Clearly, $u^* \in U$. Let $c_i = J(\theta^*; x_i, y_i)$ for $i = 1, \ldots, N$. In the proof of Theorem 3.1, it was shown that $u \in U$ solves equation 3 if and only if $\lambda = c_{min} + \frac{\gamma}{2}$ satisfies the conditions of equation 5 with respect to $u_i*$ for all $i$. Therefore, with $S_1 = S_2 = \emptyset$, Theorem 3.1 indicates that $u^* \in U_{S_1,S)_2}^*$ and hence must solve equation 3. $\square$

**Proposition 3.1.** *Let $\epsilon > 0$, and suppose for any $\theta$, $\max_{(x,y) \in \mathcal{Z}} |J(\theta; x, y)| < \infty$. Then there exists $\kappa \geq 0$ such that for any $\theta$, the following problem*

$$v_N^{MIX}(\theta) := \min_{u_1, \ldots, u_N} \sum_{i=1}^N (\frac{1}{N} + u_i) \cdot J(\theta; x_i, y_i) + \gamma_\theta \|u\|_1$$

$$\text{s.t. } u_i + \frac{1}{N} \geq 0 \quad i = 1, \ldots, N$$

*satisfies $v_N(\theta) + \frac{\kappa}{N} \geq v_N^{MIX}(\theta) \geq v_N(\theta)$. In particular, $-\gamma_\theta \leq \min_i J(\theta; x_i, y_i)$, and, for any $u^*$ solving $v_N^{MIX}(\theta)$, it holds that $u_i^* = -\frac{1}{N}$ for $i$ such that $J(\theta; x_i, y_i) > \gamma_\theta$.*

*Proof.* Fix $\theta$. Then for any $z = (x, y) \in \mathcal{Z}$, the function $\ell(\cdot, z, \theta)$ is linear, and hence Lipschitz with constant $\ell(1, z, \theta) = J(\theta; x, y) \leq \max_{(x,y) \in \mathcal{Z}} |J(\theta; x, y)| < \infty$.

By Lemma 3.1 of Staib & Jegelka (2017) and/or Corollary 2 of Gao & Kleywegt (2023),

$$v_N^{MIX}(\theta) := \min_{\tilde{w}^1, \ldots, \tilde{w}^N \geq 0} \frac{1}{N} \sum_{i=1}^N \ell(\tilde{w}^i, z^i; \theta)$$

$$\text{s.t. } \frac{1}{N} \sum_{i=1}^N |\tilde{w}^i - w^i| \leq \epsilon$$

provides the stated approximation of $v(\theta)$.

Upon introducing the change of variable $u_i = \frac{\tilde{w}^i}{N} - \frac{1}{N}$, and applying a Lagrange multiplier $\gamma_\theta$ to the $\epsilon-$ budget constraint (any convex dual optimal multiplier), we recover

$$\min_{u_1, \ldots, u_N} \sum_{i=1}^N \ell(u_i + \frac{1}{N}, z^i; \theta) + \gamma_\theta \sum_{i=1}^N |u_i|$$

$$\text{s.t. } u_i + \frac{1}{N} \geq 0 \quad i = 1, \ldots, N$$

$\square$

# B   CASE OF CONSERVATIVE ESTIMATE $C' > C$

We repeated the experimental setup of Section 4.2.1 to evaluate the benefit of RRM for MSE, but now with an estimate $C' > C$ provided to RRM.

**Analysis:** Upon comparing Table 7 with Table 4, we confirm that as the estimate $C'$ is made larger and larger than $C$, conferred benefit by RRM is reduced and is generally not as high as when $C' = C$. However, despite this, RRM-wrapping still generally outperforms MSE alone.

Table 7: **CIFAR-10** Test accuracy (%) of RRM-wrapped MSE with Conservative Estimate $C'$ of $C$ on classification task .

|  |  | Contamination $C$ | | |
|---|---|---|---|---|
|  |  | 10% | 20% | 30% |
| $C'$ | $C + 10\%$ | $91.73 \pm 0.28$ | $90.60 \pm 0.40$ | $88.47 \pm 0.73$ |
|  | $C + 5\%$ | $92.55 \pm 0.15$ | $91.29 \pm 0.41$ | $89.43 \pm 0.64$ |

# C   ENHANCING (ROBUST) LOSS FUNCTION METHODOLOGIES WITH RRM

RRM can enhance the performance of common loss functions that do not account for label contamination, including categorical cross-entropy (CCE), mean absolute error (MAE), and mean squared error (MSE). In fact, the following experiments indicate RRM can even enhance those methods devised with robustness to noisy labels, such as early-learning regularization (ELR).

Table 8: **MNIST-3** Test accuracy (%) comparison of CCE, MAE, and MSE against ELR across contamination levels. RRM-wrapped accuracy results are in parentheses.

| Method | Contamination Level | | |
|---|---|---|---|
|  | 55% | 60% | 65% |
| ELR (+ RRM) | 98 (99) | 97 (98) | 82 (87) |
|  |  |  |  |
| CCE (+ RRM) | 90 (98) | 77 (96) | 46 (67) |
| MAE (+ RRM) | 98 (98) | 96 (98) | 62 (68) |
| MSE (+ RRM) | 90 (98) | 74 (97) | 45 (87) |

**Dataset:** In this experiment, we divide MNIST-3 into a training set of 18623 examples (5923 0's, 6742 1's, and 5958 2's), and 3147 testing examples.

**Architecture:** We used a simple fully-connected architecture consisting of a few layers. The first two layer consists of 320 units, and the third layer consists of 200 units. Each of these dense layers employs the ReLU activation function. The output layer consists of a 3-unit dense layer with softmax activation. In total, there are 417880 trainable parameters.

**Experiment:** Using Tensorflow 2.10 (Abadi et al., 2015), 100 epochs of training are performed for each of the baseline loss functions (ELR, CCE, MAE, and MSE) and training label contamination levels of 55%, 60%, and 65% for a total 12 baseline experiments. In order to assess the benefit of our method, the RRM algorithm is executed for each of the baseline loss functions and contamination levels. More specifically, 10 iterations of RRM are executed with $\sigma = 10$ epochs per iteration for a total of 100 epochs. For RRM, the hyperparameter settings of $\mu$ and $\gamma$ are set to 0.5 and 0.4, although auto-tuning could have been implemented as in Section 4.2. All methods employ stochastic gradient descent (SGD) with a learning rate ($\eta$) of $0.1$. The test set accuracy of each experiment is shown in Table 8, with the RRM experimental results placed in parentheses.

**Takeaway:** For every method tested, the RRM-wrapped approach confers a test set performance benefit over the baseline approach. The additional computation time incurred from the `Re-weight`$(\gamma, \mu)$ step is an average of 2.86 seconds per execution of every iteration, for a total of 28.6 additional seconds.

# D  NON-UNIFORM LABEL CONTAMINATION EXPERIMENT

Although we explicitly state the use of uniform label noise in Section 3.1, which is indeed a very common scheme in the literature, our analysis in fact did not rely on this assumption. Towards providing insight into the non-uniform case, we have repeated the experiments of Section 4.2.2 that produced Table 5, but now with non-uniform label noise. More precisely, after uniformly at random selecting $C$ percent of the training pairs, we proceed to contaminate the label $y_i$ in each pair $(x_i, y_i)$ in the following non-uniform manner, as outlined below in the transition kernel matrix of (True Label, Contaminated Label) entries. For example, if the true label $y_i = 5$, then instead of uniformly at random drawing an alternative digit $\tilde{y}_i$ from among $\{0, 1, \ldots, 9\} \setminus \{5\}$ we have

$$\tilde{y}_i = \begin{cases} 0 & w.p.\,0.051 \\ 1 & w.p.\,0.017 \\ 2 & w.p.\,0. \\ 3 & w.p.\,0.627 \end{cases}$$

Table 9: Contamination Kernel

|  |  | \multicolumn{10}{c}{Contaminated} |
|  |  | 0 | 1 | 2 | 3 | 4 | 5 | 6 | 7 | 8 | 9 |
|---|---|---|---|---|---|---|---|---|---|---|---|
| Original | 0 | 0 | 0.0769 | 0.0769 | 0.1538 | 0 | 0.0769 | 0.3846 | 0 | 0.1538 | 0.0769 |
|  | 1 | 0 | 0 | 0.3333 | 0.1111 | 0 | 0.1111 | 0.1111 | 0 | 0.3333 | 0 |
|  | 2 | 0.0968 | 0.0645 | 0 | 0.2581 | 0.0323 | 0 | 0.0968 | 0.1935 | 0.2581 | 0 |
|  | 3 | 0 | 0 | 0.1250 | 0 | 0 | 0.1250 | 0 | 0.1250 | 0.6250 | 0 |
|  | 4 | 0.1111 | 0.0370 | 0.0741 | 0.0741 | 0 | 0.0741 | 0.2222 | 0.0370 | 0.1111 | 0.2593 |
|  | 5 | 0.0508 | 0.0169 | 0 | 0.6271 | 0.0169 | 0 | 0.1525 | 0 | 0.1017 | 0.0339 |
|  | 6 | 0.2353 | 0.1765 | 0.0588 | 0.0588 | 0.0588 | 0.1765 | 0 | 0 | 0.2353 | 0 |
|  | 7 | 0.0500 | 0.2250 | 0.2000 | 0.1250 | 0 | 0 | 0 | 0 | 0.2000 | 0.2000 |
|  | 8 | 0.1071 | 0.0357 | 0.1071 | 0.3571 | 0.1071 | 0.0714 | 0.0714 | 0.1071 | 0 | 0.0357 |
|  | 9 | 0.0638 | 0.1702 | 0 | 0.2128 | 0.1702 | 0.1702 | 0.0213 | 0.0851 | 0.1064 | 0 |

These entries were generated by the confusion matrix of an imperfect MNIST classifier. The results from this new experiment confirm the performance benefits that were observed (compare to Table 5 under conditions of uniform label contamination.

Table 10: **MNIST-10** Test accuracy (%) for AT and A-RRM under different levels of contamination $C$, training perturbation $\epsilon = 1.0$, and test-set adversarial perturbation $\epsilon_{test}$.

|  |  | \multicolumn{10}{c}{$C$} |
|  |  | 0% | | 5% | | 10% | | 20% | | 30% | |
|  |  | AT | A-RRM | AT | A-RRM | AT | A-RRM | AT | A-RRM | AT | A-RRM |
|---|---|---|---|---|---|---|---|---|---|---|---|
| $\epsilon_{test}$ | 0.00 | 96.5 | **97.3** | 93.6 | **95.6** | 60.4 | **87.8** | 32.2 | **92.4** | 58.3 | **89.2** |
|  | 0.10 | 93.4 | **95.2** | 89.3 | **92.4** | 63.2 | **84.7** | 42.5 | **89.5** | 56.1 | **81.7** |
|  | 0.25 | 92.4 | **93.1** | 87.9 | **90.6** | **86.9** | 86.3 | 80.6 | **89.3** | 69.0 | **79.8** |
|  | 0.50 | **92.0** | 90.9 | **90.3** | 89.8 | **94.4** | 91.2 | **92.9** | 89.2 | **85.2** | 81.6 |
|  | 1.00 | **89.4** | 85.5 | **90.3** | 86.8 | **94.9** | 92.6 | **93.9** | 86.6 | **81.8** | 77.8 |

# E  ADDITIONAL DATA EXPERIMENTS

## E.1  TOXIC COMMENTS

**Dataset:** Toxic Comments[3] is a multi-label classification problem from JIGSAW that consists of Wikipedia comments labeled by humans for toxic behavior. Comments can be any number (including zero) of six categories: toxic, severe toxic, obscene, threat, insult, and identity hate. We convert this into a binary classification problem by treating the label as either none of the six categories or at least one of the six categories. This dataset is a public dataset used as part of the Kaggle Toxic Comment Classification Challenge.

---

[3]https://kaggle.com/competitions/jigsaw-toxic-comment-classification-challenge

**Architecture:** We use a simple model with only a single convolutional layer. A pretrained embedding from FastText is first used to map the comments into a 300 dimension embedding space, followed by a single convolutional layer with a kernel size of two with a ReLU activation layer followed by a max-pooling layer. We then apply a 36-unit dense layer, followed by a 6 unit dense layer with sigmoid activation. Binary cross-entropy is used for the loss function.

**Experiment:** We use the Toxic Comments dataset to test the efficacy of RRM on low prevalence text data. The positive (toxic) comments consist of only 3% of the data and we contaminate anywhere from 1% to 20% of the labels. There are a total of 148,000 samples, and we set aside 80% for training and 20% for test. $\sigma = 2$ with 3 iterations of the heuristic algorithm results in a total of 6 epochs, and ERM is run for a total of 6 epochs to make the results comparable. Since the data is highly imbalanced, we look at the area under the curve of the precision/recall curve to assess the performance of the models. Unsurprisingly, as the noise increase, the model performance decreases. We note that RRM outperforms ERM across all noise levels tested, though as the noise increase, the gap between RRM and ERM decreases.

Table 11: **Toxic Comments** Comparison of training and test area under the precision/recall curve for ERM and RRM at noise levels ranging from 1% to 20%.

| Method | Percentage Contaminated Training Data | | | | | |
|---|---|---|---|---|---|---|
| | 1% | 5% | 7% | 10% | 15% | 20% |
| ERM (train) | 0.2904 | 0.2006 | 0.1589 | 0.1302 | 0.1073 | 0.0920 |
| RRM (train) | 0.6875 | 0.4458 | 0.3805 | 0.3087 | 0.2438 | 0.1966 |
| ERM (test) | 0.5861 | 0.3970 | 0.3246 | 0.2550 | 0.2013 | 0.1717 |
| RRM (test) | **0.6705** | **0.4338** | **0.3619** | **0.2824** | **0.2208** | **0.1861** |

**Takeaway:** The Toxic Comment example presents another challenging classification problem, characterized by a low prevalence target class amidst label noise. Our experiments demonstrate that as the amount of label noise increases, standard methods become increasingly ineffective. However, RRM remains reasonably robust under varying degrees of label contamination. Therefore, RRM could be a valuable addition to the set of tools being developed to enhance the robustness of AI-based decision engines.

### E.2 IMDB

**Dataset (Maas et al., 2011):** A binary classification dataset consisting of 50000 movie reviews each assigned a positive or negative sentiment label. 25000 reviews are selected randomly for training and the remaining are used for testing. 25%, 30%, 40%, and 45% of the labels of the training reviews are randomly selected and swapped from positive sentiment to negative sentiment, and vice versa, to achieve four training datasets of desired levels of label contamination.

**Architecture:** Transformer architectures have achieved SOA performance on the IMDb dataset sentiment analysis task (Devlin et al., 2019; Xie et al., 2020). As such, we a adopt a reasonable transformer architecture to assess RRM. We utilize the DistilBERT (Sanh et al., 2020) architecture with low-rank adaptation (LoRA) (Hu et al., 2022) for large language models, which reduces the number of trainable weights from 67584004 to 628994. In this manner, we reduce the computational burden, while maintaining excellent sentiment analysis performance. Binary cross-entropy is employed for the loss function.

**Experiment:** Twenty percent of the training data is set aside for validation purposes. Using Pytorch 2.1.0, 30 iterations of RRM are executed, with $\sigma = 10$ epochs per iteration for a total of 300 epochs for a given hyperparameter setting. For RRM, the hyperparameter settings of $\mu$ and $\gamma$ at 0.5 and 0.4, respectively, are based on a search to optimize validation set accuracy. For contrast, we perform a comparable 300 epochs using ERM. Both ERM and RRM employ stochastic gradient descent (SGD) with a learning rate ($\eta$) of $0.001$. In Table 12 we record both the test set accuracy achieved when validation set accuracy peaks, as well as the maximum test set accuracy. At these high levels of contamination RRM consistently achieves a better maximum test set accuracy.

**Takeaway:** We demonstrate that RRM can confer benefits to the sentiment analysis classification task using pre-trained large models under conditions of high label contamination. The success of fine-tuning in LLMs depends, in large part, on access to high quality training examples. We have

Table 12: **IMDb** Test accuracy (%) for ERM and RRM under different levels of contamination. Test set accuracy at peak validation accuracy and maximum test set accuracy are recorded.

| Method | Percentage Contaminated Training Data | | | |
| --- | --- | --- | --- | --- |
| | 25% | 30% | 40% | 45% |
| ERM | *90.2*, 90.2 | 89.5, 89.6 | 86.4, 86.6 | *80.7*, 81.1 |
| RRM | 90.1, **90.4** | *90.2*, **90.4** | *88.4*, **88.7** | 76.9, **82.6** |

shown that RRM can mitigate this need by allowing effective training in scenarios of high training data contamination. As such, resource allocation dedicated to dataset curation may be lessened by the usage of RRM.

### E.3 TISSUE NECROSIS

**Dataset**: A binary classification dataset consisting of 7874 256x256-pixel hematoxylin and eosin (H&E) stained RGB images derived from (Amgad et al., 2019). The training dataset consists of 3156 images labeled non-necrotic, as well as 3156 images labeled necrotic. The training images labeled non-necrotic contain no necrosis. However, only 25% of the images labeled necrotic contain necrotic tissue. This type of label error can be expected in cases of weakly-labeled Whole Slide Imagery (WSI). Here, an expert pathologist will provide a slide-level label for a potentially massive slide consisting of gigapixels, but they lack time or resources to provide granular, segmentation-level annotations of the location of the pathology in question. Also, the diseased tissue often occupies a small portion of the WSI, with the remainder consisting of normal tissue. When the gigapixel-sized WSI is subsequently divided into sub-images of manageable size for typical machine-learning workflows, many of the sub-images will contain no disease, but will be assigned the "weak" label chosen by the expert for the WSI. The test dataset consists of 718 necrosis and 781 non-necrosis 256x256-pixel H&E images, which were also derived from (Amgad et al., 2019). For both the training and test images, (Amgad et al., 2019) provide segmentation-level necrosis annotations, so we are able to ensure a pristine test set, and, in the case of the training set, we were able to identify the contaminated images for the purpose of algorithm evaluation.

**Architecture**: Consistent with the computational histopathology literature (Petríková & Cimrák, 2023), we employ a convolutional neural network (CNN) architecture for this classification task. In particular, a ResNet-50 architecture with pre-trained ImageNet weights is harnessed. The classification head is removed and replaced with a dense layer of 512 units and ReLU activation function, followed by an output layer with a single unit using a sigmoid activation function. All weights, with the exception of the new classification head are frozen, resulting in 1050114 trainable parameters out of 24637826. Binary cross-entropy is employed for the loss function.

**Experiment:** Twenty percent of the training data is set aside for validation purposes, including hyperparameter selection. 60 iterations of RRM are executed, with $\sigma = 10$ epochs per iteration, for a total of 600 epochs for a given hyperparameter setting. For RRM, the hyperparameter settings of $\mu$ and $\gamma$ at 0.5 and 0.016, respectively, are based on a search to optimize validation set accuracy. For contrast, we perform a comparable 600 epochs using ERM. Both ERM and RRM employ stochastic gradient descent (SGD) with a learning rate ($\eta$) of 5.0 and 1.0, respectively. RRM achieves a test set accuracy at peak validation accuracy of **74.6**, and a maximum test set accuracy **77.2**, whereas ERM achieves 71.7 and 73.2, respectively. RRM appears to confer a performance benefit under this regime of weakly labeled data.

**Takeaway:** In the Tissue Necrosis example, we demonstrate that RRM also confers accuracy benefits to the necrosis identification task provided weakly labeled WSIs. Again, RRM can mitigate the need for expert-curated, detailed pathology annotations, which are costly and time-consuming to generate.

## F SUPPLEMENT TO TABLE 6 EXPERIMENTS

In this section, we supplement the experiment of Table 6 with a follow-up investigation into individual u-value trajectories.

### F.1 CLOSE-CALLS

In Table 6, it is shown that the set of samples down-weighted to zero is not precisely $\mathcal{C}$ and some (Type-1) errors were made. Upon closer examination, an interesting behavior was observed - namely, that some samples initially down-weighted to zero were later restored to nonzero weight. In other words, these samples were close to being erroneously removed from training.

We re-ran the results of our Table 6 experiment, this time tracking all u-value trajectories. We found that of the 38400 clean-labeled samples, 37058 of them were never down-weighted to zero at any time during training. Of the remaining 1342 clean-labeled samples, 626 had sample weights $\frac{1}{N} + u_i \approx 0$ upon conclusion of training, i.e, were down-weighted to zero (Type 1 errors), whereas 716 of them were **"close-calls"**, i.e., had sample weights of $\frac{1}{N} + u_i \approx 0$ at some iteration before then recovering by the conclusion of training to have a positive sample weight $\frac{1}{N} + u_i \geq \frac{1}{2N} > 0$.

We find these results interesting because the observed behavior indicates RRM can recover a border-line but ultimately learnable sample that was assigned to near-zero weight early.

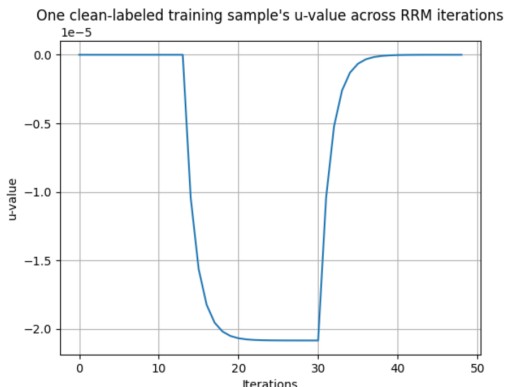
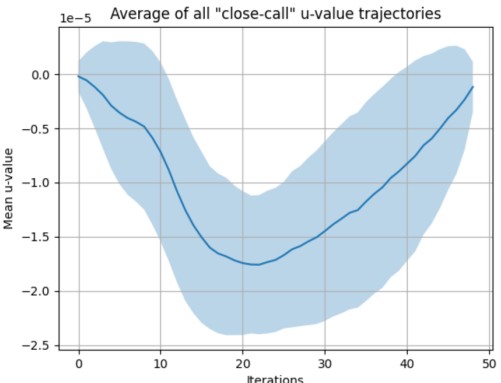

(a) The evolution of one clean-labeled training sample's $u$-value across training iterations. We note that its sample weight $\frac{1}{N} + u_i \approx 0$ at around iteration 30 before then recovering to $\frac{1}{N} + u_i \approx \frac{1}{N}$ by the conclusion of training.

(b) Average trajectory (over 716 "close-call" clean-labeled samples). Shading indicates one standard deviation.

Figure 1: Behavior of $u$-values during training.

