# OpenReview forum: "Enhancing Learning with Noisy Labels via Rockafellian Relaxation"
_ICLR.cc/2026/Conference — ICLR 2026 Poster_

### Official Review · Reviewer_yHgs · 2025-10-27

**Soundness:** 1
**Presentation:** 1
**Contribution:** 2
**Rating:** 2
**Confidence:** 4

**Summary:**

This paper introduces the Rockafellian relaxation method to enhance the capacity of neural networks to tolerate noisy labels.

**Strengths:**

- The authors make an effort to propose a general framework for improving robustness in learning under noisy labelling conditions.

- The paper attempts to provide both theoretical analysis and empirical results, showing the authors’ intention to bridge formulation and practice.

**Weaknesses:**

This submission contains a significant formatting violation that should be desk-rejected. Specifically, the appendix section appears before the bibliography, causing the main text to exceed the 9-page limit.

Regarding the work itself, the current version is very difficult to follow, and the overall presentation lacks clarity. The writing requires substantial improvement to enhance readability and logical flow.

In the next revision, the authors should at least provide a clear introduction to the concept of Rockafellian relaxation and explain the motivation for applying this technique to address label noise. The paper should also avoid overcomplicating simple concepts. For example, Equations (1) and (2) are described in an overly complex manner.

**Questions:**

NA

---

> ### Author Response · Authors · 2025-11-13
> **On motivation/clarity/formatting**
>
> - The main text concludes with the conclusion section on page 9, so the main text 9-page limit was not violated. We have reordered the appendix and references section in the latest uploaded manuscript. But we emphasize that our manuscript has, and still, obeys the main text 9-page limit.
> - As for equation (1), $\mathcal{L}(\theta):= \frac{1}{N} \sum_{i = 1}^N  J(\theta; x_i,y_i) + r(\theta)$, this is a textbook expression for (regularized) deep learning loss that is ubiquitously employed in the literature - see, for example, equations (7.1), (8.1), (8.2), and (8.3) in Deep Learning by Ian Goodfellow, Yoshua Bengio & Aaron Courville. Given the standard nature of this equation, we believe our short description of this equation is succinct without loss of clarity and no more complex than it had to be.
> - As equation (2) is integral to our paper, we provided both a concise, mathematically formal description (lines 116- 127) as well as a more elaborated, natural language description (lines 128-133, passage beginning with "In words,..."). We'll try to explain again here less formally: in equation (2), we depart from equation (1) by now allowing for the $(x_i, y_i)$  pairs to be distributed not by the empirical $p^N$ distribution, but by an alternatively selected one, $p$, that will help in minimizing loss; however, if this selection is not kept in check, then $p$ would trivially be the distribution that focuses all mass on the pair $(x_i, y_i)$ whose loss $J(\theta; x_i, y_i)$ is smallest, hence the selection should come at a cost, for which we devised as $\gamma$ times the total variation distance between $p$ and $p^N$ (equivalently, $\frac{\gamma}{2}  \|\|p-p^N\|\|_1$).
> - As for the **motivation** behind this technique, we kindly refer the reviewer to the passage immediately preceding mention of the Rockafellian Relaxation Method, which reads: *"However, if we have noisily-labeled data, we might desire to remove those members that are affected; in other words, if $\mathcal{C} \subsetneq \\{1, ... , N\\}$ is the set of contaminated observations, then we might desire instead to have $p_i = 0$ when $i \in \mathcal{C}$, and $p_i = \frac{1}{N - |\mathcal{C}|}$ otherwise. In this work, we consider modification of those methods that optimize a loss $\mathcal{L}$ resembling (1), with the intention of aligning the $p_i$ values closer to these desired (but unknown) $p$. "* **In short**, our motivation is to deal with noisily-labeled (contaminated) training observations by altering the weightings (probability masses) given to them.
> - As for your request that the manuscript contain an introduction to the concept of Rockafellian Relaxations...this is not critical to understanding equation (2), the principle idea of our paper; indeed, Rockafellian Relaxations assume an accessible form (for machine learning readers) in this context of noisy labels, but takes on different and less relevant interpretations outside this context, which would be out of scope. Considering this and the page limit, we felt it was most economical to omit an introduction to this concept but we referred the interested reader to literature.

---

### Official Review · Reviewer_myha · 2025-10-29

**Soundness:** 1
**Presentation:** 2
**Contribution:** 2
**Rating:** 6
**Confidence:** 3

**Summary:**

The paper proposes the Rockafellian Relaxation Method (RRM), a simple, architecture-agnostic loss reweighting scheme for learning with noisy labels. RRM formulates training as a min-min problem over model parameters and example weights with a TV-style penalty, yielding a linear program with a closed-form solution that prunes high-loss samples beyond a threshold cmin + γ and redistributes their mass to lowest-loss samples. The method can be auto-tuned via an estimated contamination rate C′ and can be used with or without adversarial training (A-RRM). Experiments on CIFAR-N, Clothing1M, Food-101N, CIFAR-10, MNIST, IMDb, Toxic Comments, and histopathology show improvements over baselines and, in some cases, state-of-the-art results.

**Strengths:**

1. Simple, general “wrapper” that can enhance diverse methods and losses without architectural changes.
2. Clear optimization view with an explicit, efficient solution; connection to optimistic DRO/Wasserstein formulations is insightful.
3. Auto-tuning strategy to control pruning via an estimate of noise level.
4. Broad experimental sweep (vision, NLP, medical imaging) with competitive or improved results; SOTA on CIFAR-100N when wrapping strong baselines.
5. Useful analysis of u’s convergence showing selective downweighting of noisy points; practical relevance when test-time perturbations differ from training.

**Weaknesses:**

1. Novelty is moderate: the resulting rule effectively implements trimmed/quantile ERM focused on smallest-loss samples; closely related to self-paced/trimmed risk/reweighting/DRO literature (e.g., self-paced learning, MentorNet, trimmed loss), but comparisons to these specific baselines are missing.
2. Dependence on contamination estimate C′: despite some ablations, guidance for estimating C′ without clean data is thin; sensitivity analysis is limited.
3. Experimental fairness and reporting: some baselines lack confidence intervals; training schedules and adversarial settings are not always matched; improvements are sometimes small or inconsistent (e.g., MAE).
4. The method of weighting samples is widely used in the field of noisy label learning, such as [1][2]. However, the authors did not conduct a comparison. Furthermore, the baseline used by the author is quite outdated. It would be better to incorporate some more recent related works and baselines, such as those from 2024 and 2025.
5. The author should include a comparison of the time spent. I think this is very valuable for reference.

[1] Knockoffs-SPR: Clean Sample Selection in  Learning with Noisy Labels.

[2] Meta-Weight-Net: Learning an Explicit Mapping For Sample Weighting.

**Questions:**

Please see Weaknesses.

---

> ### Author Response · Authors · 2025-11-21
> **Answers to Weaknesses/Questions 1,2,3**
>
> **Weakness/Question 1:** "*Novelty is moderate: the resulting rule effectively implements trimmed/quantile ERM focused on smallest-loss samples; closely related to self-paced/trimmed risk/reweighting/DRO literature (e.g., self-paced learning, MentorNet, trimmed loss), but comparisons to these specific baselines are missing.*"
> **Answer:**
> The noisy-label literature indeed has many methods that we could compare against, including the ones you mentioned: self-paced learning, MentorNet, and trimmed loss. With so many methods to choose from, we strategically focused on those methods that were reported to have the best accuracies on real-world contamination settings - namely CIFAR-N, Clothing1M, and Food-101N. Unfortunately, self-paced learning, MentorNet, and trimmed loss did not appear (near the) top of the leaderboards. For what it's worth, we greatly outperformed the baseline *Co-Teaching +* , which is a method devised to improve upon self-paced curriculum learners like MentorNet - see *Co-teaching: Robust Training of Deep Neural Networks with Extremely Noisy Labels* for discussion on this.
>
> **Weakness/Question 2:** "*Dependence on contamination estimate C′: despite some ablations, guidance for estimating C′ without clean data is thin; sensitivity analysis is limited.*"
> **Answer:**
> As far as a sensitivity analysis, we kindly refer to Table 7 of Appendix Section B wherein we performed synthetic experiments to examine what happens to performance when the estimated level of contamination $C'$ departs from the actual contamination level $C$. In the experiment of Table 7, $C'$ exceeding $C$ yielded relatively minimal dropoffs in performance.
>
> **Weakness/Question 3:** "*Experimental fairness and reporting: some baselines lack confidence intervals; training schedules and adversarial settings are not always matched; improvements are sometimes small or inconsistent (e.g., MAE).*"
> **Answer:**
>
> **On "some baselines lack confidence intervals"**
> - In the LNL (Learning with noisy labels) literature, it is in fact common to not report confidence intervals. For example, the references [1], [2] you provided did not include confidence intervals for any of the large datasets like Clothing1M and WebVision. For examples of other top papers doing this, we refer to: "ProMix: Combating Label Noise via Maximizing Clean Sample Utility", "DivideMix: learning with noisy labels as semi-supervised learning", and "Centrality and Consistency: Two-Stage Clean Samples Identification for Learning with Instance-Dependent Noisy Labels". In truth, many LNL papers explain their omission as a matter of computational expense. For what it's worth, we reported standard deviations in a fair number of our results, as compared to many papers in the literature that will omit this entirely.
> - In lines 370-371, we wrote: *"In Tables 1,2, and 3, wherever confidence intervals are omitted, this is because they are in fact not reported by the authors of that method."*
>
> **On "training schedules and adversarial settings are not always matched"**
> We are not entirely clear what is "not always matched", but  our best guess is that this is referring to hyperparameter setting/selection. We briefly comment on why sometimes there may have been similar versus different hyperparameter settings/selections.
> - On Line 334, we write: *"As for any hyperparameter of ProMix, Divide-Mix, or CC, in our experiments we simply re-used those reported in their respective papers’ (Xiao et al., 2023; Li et al., 2020; Zhao et al., 2022) experimental setup."*
> - Hyper-parameter selection is based on the validation set, and one should not expect it to remain consistent across different experimental datasets.
>
> **On "improvements are sometimes small or inconsistent (e.g., MAE)"**
> - We reiterate that all methods we wrapped outside of MAE and GCE produced elevated test accuracies (or did no harm). Further, this was observed across myriad data regimes, architectures, and tasks (we counted 36 experiments).
> - We don't argue that RRM will benefit every wrapped method greatly for every dataset, but rather that it can and sometimes does achieve state-of-the-art performance.

---

> ### Author Response · Authors · 2025-11-21
> **Answers to Weaknesses/Questions 4,5**
>
> **Weakness/Question 4:** "*The method of weighting samples is widely used in the field of noisy label learning, such as [1][2]. However, the authors did not conduct a comparison. Furthermore, the baseline used by the author is quite outdated. It would be better to incorporate some more recent related works and baselines, such as those from 2024 and 2025.*"
> **Answer:**
> We thank you for providing these two additional baselines for comparison. In Section 4.1, we selected baselines that were: (1) tested on real-world contamination settings - namely CIFAR-N, Clothing1M, and Food-101N; (2) atop the leaderboards for these datasets. In valuing these criteria, we may have neglected some more recent baselines. Nonetheless, your references (Knockoffs-SPR and Meta-Weight-Net) have been added to our Table 2's comparison of top performers for Clothing1M (they had respective performances of 75.2 and 73.72), for which our reported 75.69 maintains its position. Our latest submitted revision contains this addition as well as their citations.
>
> **Weakness/Question 5:** "*The author should include a comparison of the time spent. I think this is very valuable for reference.*"
> **Answer:**
> For a clear understanding of how to compare "time spent", it may help to clarify where RRM-wrapping inserts into typical training procedure so as to highlight where additional time is incurred.
>
> Quite simply, RRM-wrapping inserts a single step into standard neural network training procedures. More precisely, after each iteration's "standard training", e.g., SGD steps, has concluded, there is a call to the procedure $Re$-$weight(\gamma, \mu)$- see Algorithm 1 and Algorithm 3. Therefore,
>
> $$(\text{RRM-wrapped Training Time}) = (\text{Unwrapped Training Time}) +\underset{RRM-overhead}{\underbrace{ \text{(Time dedicated to executing $Re$ - $weight(\gamma, \mu)$ )}}} $$
>
> Hence, for comparison between wrapped and unwrapped methods, the key distinction will lie with the total time spent on loss reweighting, which is conducted by solving the linear program of equation (3) either with a solver or via Corollary 3.1.1 which requires just a single pass over the list of losses. For more details on this **lightweight** step, we kindly refer to Section 3.5.3 of our paper.
>
> **Reported Time spent:**
> RRM-wrapped training time will clearly vary according to which method we're wrapping with RRM (as well as the dataset and computing power). Our experiments of Sec. 4 typically conducted a reweighting step approximately once every  $\sigma=10$ epochs. We reported the average computation time for a single reweighting step as 3.88 seconds (see lines 395-396 of paper) for the experiments of Sec. 4.2.1 and 2.86 seconds for the experiments in Appendix C. **In conclusion, we appreciate the opportunity to provide clarification on this matter. All our experiments found negligible overhead that was insignificant with respect to the total training times, so further discussion and comparison beyond the above reporting was not provided.**

---

> > ### Comment · Reviewer_myha · 2025-11-25
> >
> > The author's response addressed most of the questions I raised, and I maintained my positive score.

---

### Official Review · Reviewer_XbZp · 2025-10-30

**Soundness:** 4
**Presentation:** 3
**Contribution:** 3
**Rating:** 8
**Confidence:** 4

**Summary:**

The authors propose Rockafellian Relaxation (RRM), a method for learning with noisy labels that jointly optimizes model parameters $\theta$ and per-sample weights $u$ under a total-variation style constraint. For fixed $\theta$, RRM solves a linear program to obtain weights $(1/N + u_i)$ on each training example $i$, minimizing $\sum(1/N + u_i) \cdot J(\theta; x_i, y_i)$ $+ (\gamma / 2) \lvert u \rvert_{1}$ subject to $1/N + u_i \geq 0$ and $\sum u_i = 0$. The optimal $u$ places nearly zero weight on high-loss samples whose supervised loss exceeds $c_{min} + \gamma$ and reallocates their probability mass to the lowest-loss set. Training alternates between: (i) taking $\sigma$ steps of SGD on $\theta$ using these per-sample weights, and (ii) re-solving the linear program using updated losses to recompute $u$. The authors present the method as a wrapper for existing noisy-label pipelines such as DivideMix, ProMix, and CC. They report accuracy gains on CIFAR-10N/100N, Clothing1M, Food-101N, and two NLP datasets.

**Strengths:**

1. The optimization view of noisy-label learning is compelling. RRM reframes sample selection as an explicit joint optimization over $\theta$ and a perturbed empirical distribution, rather than as a heuristic small-loss filter. The inner reweighting problem is convex in $u$ for fixed $\theta$ and exhibits a closed-form thresholding structure: all samples above $c_{min} + \gamma$ are effectively suppressed, and their mass is reassigned to the minimum-loss set.
2. The method seems lightweight and practical. The algorithm alternates two inexpensive steps ($\sigma$ steps of weighted SGD, then one reweight solve), and the contamination prior $C'$ optionally sets $\gamma$ through a loss quantile so that pruning aggressiveness is tied directly to an estimated corruption rate.
3. The method functions as a wrapper around already strong noisy-label learners that combine supervised and semi-supervised components. The authors convincingly demonstrate consistent improvements on realistic label noise benchmarks (CIFAR-N, Clothing1M, Food-101N).

**Weaknesses:**

1. The integration of RRM with semi-supervised pipelines is somewhat underspecified.
    * The authors model existing noisy label methods as having a supervised term $J(\theta; x_i, y_i)$ and an additional term $r(\theta)$, then state that they replace the training loss with $L_{RRM}$ (i.e., $\theta$ updates weight each sample's gradient by $(1/N + u_i)$).
    * What is not made explicit is how $r(\theta)$ participates in those $\theta$ updates once RRM is active. Many noisy label methods (e.g., DivideMix, ProMix, CC) employ two-branch training: one branch treats some samples as labeled and optimizes a supervised loss on them, while the other uses the remaining samples in a semi-supervised objective (consistency, pseudo-labeling, auxiliary regularizers).
    * In RRM, samples whose current supervised loss $J(\theta; x_i, y_i)$ exceeds $c_{min} + \gamma$ can be assigned weight $(1/N + u_i) \approx 0$, which zeroes their gradient contribution in the weighted SGD phase. The critical question is whether these samples still contribute through the semi-supervised branch $r(\theta)$ as unlabeled/consistency signal, or whether they are effectively removed from training altogether.
    * I point this out because methods like DivideMix are designed so that even suspected-noisy samples influence training through their semi-supervised branch rather than being discarded. If RRM can fully suppress those samples in the supervised branch when RRM and the inner wrapped method disagree on the noisy/clean partition, the optimization may not be maximally using all of the available data.
2. The authors do not fully characterize the trajectory of sample weights over time.
    * The authors report that the set of pruned samples converges and stabilizes during training.
    * However, to my knowledge, the paper does not show trajectories of individual samples. Interesting behavior might arise: for example, some samples initially downweighted to $(1/N + u_i) \approx 0$ may or may not later be restored to nonzero weight. Such behavior would show whether the method can recover a borderline but ultimately learnable sample that was assigned to near-zero weight early.

**Questions:**

1. When a sample is assigned $(1/N + u_i) \approx 0$ because its supervised loss exceeds $c_{min} + \gamma$, do you continue to use that sample in the semi-supervised component $r(\theta)$ (for example, as unlabeled data in a consistency or pseudo-label objective), or is it excluded from training until it regains positive weight?
2. During $\theta$-updates in your wrapped DivideMix/ProMix/CC runs, are gradients from $r(\theta)$ also scaled by $(1/N + u_i)$, or are they left unweighted?
3. More broadly, what representational insight does RRM offer that is of direct interest to the ICLR community? How does this work advance our understanding of learned representations, beyond improving robustness to noisy labels?

I would be happy to further increase my score if the authors can answer these questions and revise the manuscript accordingly.

---

> ### Author Response · Authors · 2025-11-19
> **Answer to Weakness 1**
>
> We appreciate the opportunity to explain here in more detail how we integrated RRM with semi-supervised pipelines as in DivideMix, ProMix, and CC. We start with a high-level explanation before getting into how we specifically executed the wrapping of each of the three.
>
> **High level explanation:**
>
> Roughly speaking, ProMix, Divide-Mix, and CC all execute some procedure to divide the labeled training data
> $D = \\{(x_i, y_i)\\}$
> into a set $X$ for supervised training, and a set $U$ for semi-supervised training. This results in a combined loss of the form
> $L(\theta) := L_X(\theta) + r(\theta)$,
> where
> $r(\theta) := \lambda_U L_U(\theta) + L_{aux}(\theta)$
> collects a ($\lambda_U$-weighted) semi-supervised component $L_U$, as well as any additional auxiliary loss component $L_{aux}$. In fact, the supervised set $X$ and semi-supervised set $U$ are re-derived in each epoch, so any particular sample $(x_i, y_i)$ may switch between being in $X$ and $U$ across the epochs of training. In any case, given an $X$, the resulting supervised component of loss always assumes the form
> $L_X(\theta) = \frac{1}{|X|} \sum_{i \in X} J(\theta; x_i, y_i)$
> for some classification loss $J$. Therefore, when we say we wrapped one of these methods with RRM, roughly speaking, we mean that we leave untouched the $X$ and $U$ derivation procedures, but we do replace $L_X(\theta)$ with the $L_{RRM}(\theta)$ of our equation (2), wherein $N$ is now the cardinality $|X|$ of a supervised set $X$. Below, we provide detailed explanations for each method.
>
> **Detailed Wrapping Specifics:**
>
> **Divide-Mix:** We made the following amendments to Algorithm 1 of *"DIVIDEMIX: LEARNING WITH NOISY LABELS AS SEMI-SUPERVISED LEARNING"*:
>
> - Introduced a collection of global $u_i$ variables, one for each member of the training data, all initialized to 0.
>
> - If we are in an epoch in which no $u$-update is to be performed, then in line 25 we set  $L_X(\theta) = \sum_{i \in X'} \left(\frac{1}{B \cdot M} + u_i\right) J(\theta; x_i, y_i),$ where $B$ is batch size and $M$ is the number of augmentations performed.
>
> - If we are in an epoch in which a $u$-update is to be performed, then after the SGD step of line 27 we reset the global $u_i$ variables:
>   - $u_i$ unchanged for all $i \notin X'$
>   - for the $u_i$ in which $i \in X'$, these are set to a solution of $\min_{u \in U} \sum_{i \in X'} \left(\frac{1}{B \cdot M} + u_i\right) J(\theta; x_i, y_i) + \frac{\gamma}{2} \|\|u\|\|_1$  (via Corollary 3.1.1 of our paper)
>
>
> **CC:** As outlined in Section 3.4 of *"Centrality and Consistency: Two-Stage Clean Samples Identification for Learning with Instance-Dependent Noisy Labels"*, the method CC's last step in any epoch is to formulate $L_X$ and $L_U$ in exactly the same manner as Divide-Mix. Therefore, we applied the same edits described above for Divide-Mix.
>
> **ProMix:** We made the following amendments to Algorithm 1 of *"ProMix: Combating Label Noise via Maximizing Clean Sample Utility"*:
>
> - Introduced a collection of global $u_i$ variables, one for each member of the training data, all initialized to 0.
>
> - If we are in an epoch in which no $u$-update is to be performed, then in line 22 the original head component
>   $L_X(y_i, p_i)$  is set to  $\sum_{i \in D_l} \left(\frac{1}{|D_l|} + u_i\right) H(\tilde{y}_i, p_i).$
>
> - If we are in an epoch in which a $u$-update is to be performed, then after the minimization step of line 26 we reset the global $u_i$ variables:
>   - $(u_i)_{i \notin D_l}$ unaltered
>   - for the $u_i$ in which $i \in D_l$, these are set to a solution of  $\min_{u \in U} \sum_{i \in D_l} \left(\frac{1}{|D_l|} + u_i\right) H(\tilde{y}_i, p_i) + \frac{\gamma}{2} \|\|u\|\|_1$  (via Corollary 3.1.1 of our paper)
>
>
> **Bottom Line:**
>
> If a sample $i$'s gradient contribution is zeroed (e.g., $(1/N) + u_i \approx 0$) in an epoch, then $i$ is part of the supervised component ($i \in X$) in that epoch, and then because $X \cap U = \emptyset$, this must mean $i \notin U$ and hence it cannot simultaneously contribute through the semi-supervised branch in that same epoch. However, this does not preclude the possibility of sample $i$ being selected to be part of $U$ and contributing through the semi-supervised branch in a later epoch, just as this could happen in Divide-Mix. Therefore, it's not the case that zeroed-out samples in the supervised branch become fully suppressed or discarded from ever contributing again during the training process.

---

> ### Author Response · Authors · 2025-11-19
> **Answer to Weakness 2**
>
> You're correct that although we discuss/comment and show (via Table 6) the trajectory of the u-values as a collective we do not focus on any one particular sample's u-value trajectory. Considering there were $48,000$ u-values in Table 6, we didn't think to identify/focus on any one particular sample. However, your comment has since inspired us to identify those samples $i$ among those with clean labels, i.e., $i \notin C$, and whose u-value $u_i$ was down-weighted to zero at some time during the training; in particular, we investigated your query: *did these samples have u-values that were or were not later restored to nonzero weight?* Indeed, we really appreciated your comment that, *"Such behavior would show whether the method can recover a borderline but ultimately learnable sample that was assigned to near-zero weight early.*"
>
> **Results of our follow-up investigation:**
> We re-ran the results of our Table 6 experiment, this time tracking all u-value trajectories. We found that of the 38400 clean-labeled samples, 37058 of them were never down-weighted to zero at any time during training. Of the remaining 1342 clean-labeled samples, 626 had sample weights $\frac{1}{N} + u_i \approx 0$ upon conclusion of training, i.e, were down-weighted to zero (Type 1 errors), whereas 716 of them had sample weights of $\frac{1}{N} + u_i \approx 0$ at some iteration before then recovering by the conclusion of training to have a positive sample weight $\frac{1}{N} + u_i \geq \frac{1}{2N} > 0$.
>
> We have added two plots to a new Appendix Section F in the most recently updated manuscript.
> - In Figure 1 of Appendix Section F, we plot the evolution of one clean-labeled training sample's u-value across training iterations. We note that its sample weight $\frac{1}{N} + u_i \approx 0$ at around iteration 30 before then recovering to $\frac{1}{N} + u_i \approx \frac{1}{N}$ by the conclusion of training.
> - In Figure 2 of Appendix Section F, we identified all "close-call" clean-labeled samples, i.e., those which had sample weights of $\frac{1}{N} + u_i \approx 0$ at some iteration before then recovering by the conclusion of training to have a positive sample weight $\frac{1}{N} + u_i \geq \frac{1}{2N} > 0.$ The average (over these 716 "close-calls") u-value trajectory is plotted, with shading indicating one standard deviation.

---

> ### Author Response · Authors · 2025-11-19
> **Answers to Questions**
>
> # Answer to Question 1:
>
> ### Question:
> "*When a sample is assigned $\frac{1}{N} + u_i \approx 0$ because its supervised loss exceeds $c_{min} + \gamma$ , do you continue to use that sample in the semi-supervised component $r(\theta)$ (for example, as unlabeled data in a consistency or pseudo-label objective), or is it excluded from training until it regains positive weight?*"
>
> ### Answer:
> Since the RRM-reweighting action is only applied to supervised components, if in epoch $e,$ sample $i$ has been down-weighted to zero, then $i$ must be involved in the supervised component in that epoch of Divide-Mix/CC/Pro-Mix and hence won't contribute to the semi-supervised component **in epoch e.** However, sample $i$ could in a later epoch $e' > e$ be assigned to the semi-supervised component, and in which case, even if its u-value $u_i = -1/N$, sample $i$ would participate in the $\mathcal{L}_\mathcal{U}$ loss (and not $\mathcal{L}_X$) in epoch $e'$, just as dictated in Divide-Mix/CC/Pro-Mix.
> # Answer to Question 2:
> ### Question:
> *"During $\theta$ updates in your wrapped DivideMix/ProMix/CC runs, are gradients from $r(\theta)$ also scaled by $(\frac{1}{N} + u_i)$, or are they left unweighted?"*
>
> ### Answer:
> They are left unweighted. More precisely, during a $\theta$ update, all samples $i$ who have been determined to take part in the $\mathcal{L}_\mathcal{U}$ loss component of $r(\theta)$ will do so just as in DivideMix/ProMix/CC, that is, no u-value intervention.
> # Answer to Question 3:
> ### Question:
> *"More broadly, what representational insight does RRM offer that is of direct interest to the ICLR community? How does this work advance our understanding of learned representations, beyond improving robustness to noisy labels?"*
>
> ### Answer:
> **Representational insight**
> Whereas some representational learning approaches like autoencoding effectively seek a suitable compressed (equiv. lower dimensional) representation of the data space, RRM seeks to find a representation in a higher dimensional space. Indeed, when learning with noisy labels, it may be useful to append to every (feature, label) pair another dimensional context to help determine the legitimacy of that pairing. More precisely, for each sample $(x_i, y_i)$, could be an associated $w_i$ that expresses some kind of confidence in the noisiness of the provided label $y_i$ for $x_i$. In other words, we think RRM provides an interesting approach that runs somewhat contrary to some standard representation approaches.
>
> **Advance Understanding**
> We think RRM's use of optimistic optimization for learning a representation is a novel concept and one that makes sense for the context of learning with noisy labels. From Section 3.4 of our paper, RRM approximately approaches the task of learning with noisy labels by: (1) effectively lifting the (feature $x$, label $y$) - data space $\mathcal{X} \times \mathcal{Y}$ into a higher dimensional space (confidence $w$, feature $x$, label $y$)- space $\mathcal{W} \times \mathcal{X} \times \mathcal{Y}$; (2)  estimate a distribution over this higher dimensional space via optimistic optimization of loss. At a high-level, the intuition is that noisy (as well as adversarial) labels will tend to elevate training cost, so that choices of distributions (departing from the empirical) that optimistically lower training cost can provide a way to counteract this influence, and in doing so, also provide a means to remove some noisily-labeled samples from training.

---

> ### Comment · Reviewer_XbZp · 2025-11-26
>
> I thank the authors for their detailed responses and additional experiments.
>
> 1. **Regarding Question 1:** The clarification about how RRM interacts with the semi-supervised component is clear, and my confusion is fully resolved.
>
> 2. **Regarding Question 2:** I am glad to see that you found the suggestion interesting enough to run the “close-call” analysis. The results you report are convincing. However, the description of this experiment in Appendix F is currently hard to follow for a reader who has not seen this discussion thread. I encourage you to revise that section so that a first-time reader can easily understand (i) what “close-calls” are, (ii) why they are interesting, and (iii) how the trajectories were constructed and analyzed.
>
> 3. **Regarding Question 3:** I appreciate the attempt to situate RRM relative to representation-motivated approaches. That said, I remain unconvinced that the current framing adequately justifies the paper as a contribution to the “learning representations” theme. In particular, the revised manuscript does not yet adopt this perspective in a sustained way. I would very much appreciate it if the camera-ready could expand the related work and discussion sections to more directly engage with the literature on representation learning (e.g., autoencoding and related approaches), and to articulate more clearly how RRM offers a complementary view.
>
> I believe my overall assessment is still consistent with these comments, so I will keep my score as-is.

---

### Author Response · Authors · 2025-12-02
**Summary of Rebuttal/Discussion**

We thank the review team for their feedback and engagement during the rebuttal/discussion period. As the period closes, we summarize/highlight key points raised/discussed.

**XbZp**:
- **We thank the reviewer for their positive score (8)**
	- Reviewer characterized our optimization view as "compelling", called our method "lightweight and practical", and states that we "convincingly demonstrate consistent improvements on realistic label noise benchmarks".
- We thank the reviewer for their questions, for which we provided answers and additional experiments. The reviewer kept their score "as-is".
	- We clarified how RRM interacts with semi-supervised pipelines, and Reviewer XbZp replied that their "confusion is fully resolved".
	- Reviewer's inquiry into trajectories of individual samples, specifically those that are "borderline but ultimately learnable", i.e., "close-calls," led to us running follow-up experiments on "close-calls," adding 2 plots to our Appendix F. Reviewer XbZp found these follow-up results "convincing," and asked us to improve the description of the experiment in Appendix F. We have since done so, and thank the reviewer for their kind suggestions.
	- Reviewer requests that "the camera-ready $\ldots$ expand the related work and discussion sections to more directly engage with the literature on representation learning $\ldots$"

**myha**:
- **We thank the reviewer for their positive score (6).**
	- Reviewer notes that our method is "simple", "general", and can enhance "diverse methods", as well as characterizing our connection to optimistic DRO/Wasserstein formulations as "insightful".
	- Reviewer notes our broad experimentation with "competitive or improved results"
	- Reviewer calls our analysis of $u$'s convergence "useful" and having "practical relevance".
- We thank the reviewer for their questions, for which our answers "addressed most of the questions [they] raised, and [they] maintained [their] positive score."
	- We commented on additional baselines that the Reviewer inquired about
	- In response to the Reviewer's comment that sensitivity analysis was "limited," we referred to the relevant Appendix section in case the Reviewer had only considered the main text.
	- We clarified/re-emphasized for the reviewer that wherever confidence intervals were omitted, this is because they are in fact not reported by the authors of that method.
	- We clarified for the reviewer that the hyperparameters used in the wrappings were simply those that were reported in their respective papers.
	- We reiterated that all methods wrapped outside of MAE and GCE produced elevated test accuracies (or did no harm) in our experimentation across myriad data regimes, architectures, and tasks ($\approx 36$ experiments in total).

**yHgs**:
- Reviewer found the manuscript "difficult to follow" and claimed that the presentation "lacks clarity." Specifically, the reviewer had trouble with concepts as early in the paper as Equations (1) and (2), as well as the motivation behind our use of the Rockafellian Relaxation to address label noise.
    - We re-explained the concepts behind Equations (1) and (2). We re-explained the motivation behind the Rockafellian Relaxation.
- There was no engagement with our rebuttal by this reviewer (even before reviewer commenting was disabled). We assume our clarifications and references to relevant passages in the manuscript were sufficient.

---

### Meta-Review · Area_Chair_5ix9 · 2026-01-01

**Summary:**

The paper proposes the Rockafellian Relaxation Method (RRM) -- an architecture-independent, loss reweighting approach to enhance the capacity of neural network methods to accommodate noisy labeled data.

Based on feedback from most reviewers, this paper offers a compelling optimization perspective on noisy-label learning. The proposed method is lightweight and practical, and the convergence analysis—which demonstrates selective downweighting of noisy points—is considered valuable.

While a reviewer noted that the paper is challenging to read, considering the overall scores from all reviewers, I recommend acceptance of this submission.

**Reviewer Concerns:**

As the first two reviewers themselves have indicated, their concerns have largely been addressed. Regarding the third reviewer (yHgs), his/her comments are relatively superficial and thus should not serve as the primary basis for determining whether to accept the paper.

**Reviewer Scores:**

The first two reviewers have provided very positive feedback and have explicitly stated that they will not change their positive scores. The third reviewer did not participate in the discussion and therefore is unlikely to modify their score either.

---

### Decision · Program_Chairs · 2026-01-26

Accept (Poster)